# TabDPT: Scaling Tabular Foundation Models on Real Data

**Junwei Ma**[*], **Valentin Thomas**[*,1], **Rasa Hosseinzadeh, Alex Labach, Hamidreza Kamkari,**
**Jesse C. Cresswell, Keyvan Golestan, Guangwei Yu, Anthony L. Caterini**[1], **Maksims Volkovs**
Layer 6 AI, Toronto

## Abstract

Tabular data is one of the most ubiquitous sources of information worldwide, spanning a wide variety of domains. This inherent heterogeneity has slowed the development of Tabular Foundation Models (TFMs) capable of fast generalization to unseen datasets. In-Context Learning (ICL) has recently emerged as a promising solution for TFMs, enabling dynamic adaptation to new tasks without additional tuning. While many studies have attempted to re-purpose large language models for tabular ICL, they have had limited success, so recent works have focused on developing tabular-specific foundation models. In this work, we propose an approach to combine ICL-based retrieval with self supervised learning to train tabular foundation models. We also investigate the utility of real vs. synthetic data for model pre-training, and show that real data can contain useful signal not easily captured in synthetic training. Specifically, we show that incorporating real data during the pre-training phase can lead to significantly faster training and better downstream generalization to unseen data. Our resulting model, **TabDPT**, achieves strong performance on both regression (CTR23) and classification (CC18) benchmarks. Importantly, we also demonstrate that with our pre-training procedure, scaling both model and data size leads to consistent performance improvements that follow power laws. This echoes scaling laws in LLMs and other foundation models, and suggests that large-scale TFMs can be achievable. We open-source our full pipeline: inference code including trained model weights can be found here, and the training code to reproduce experiments can be found here.

## 1 Introduction

Tabular data constitutes the backbone of most real-world applications, from finance, healthcare, and e-commerce, to many others [59]. However, building a *Tabular Foundation Model* (TFM) – a single model that generalizes across the enormous heterogeneity of tabular datasets – remains a key challenge. The alternative, traditional approach [9, 50] is to train individual models for each new task, which may yield strong results but requires costly model selection and hyperparameter tuning on a per-dataset basis. For deep learning methods, this procedure results in even greater computational overhead,

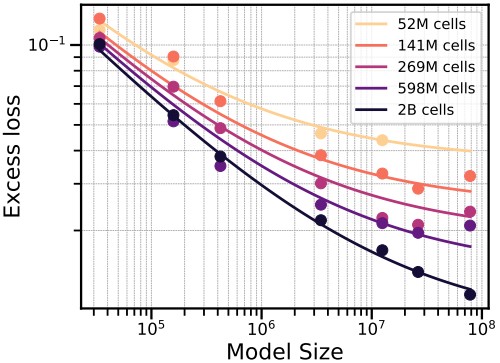

Figure 1: Scaling behavior for our foundation tabular models. Increasing model or pre-training data size (number of cells) leads to consistent improvements predictable by power laws (fitted solid lines).

which has hindered the adoption of neural networks as a universal solution in the tabular domain.

[*]Equal Contribution; [1]Correspondence to {`valentin.t, anthony`}@layer6.ai

39th Conference on Neural Information Processing Systems (NeurIPS 2025).

In-context learning (ICL) offers an appealing alternative by enabling a model to adapt to new tasks by simply modifying the context, obviating the need for per-dataset fine-tuning or hyperparameter selection [8]. Beyond lowering the cost of model deployment, ICL facilitates rapid prototyping and provides a natural mechanism for handling distribution shifts, as the model can efficiently adapt to new data at inference time using only in-context examples, mimicking the effect of conventional training with less overhead.

Recent attempts to repurpose large language models (LLMs) for ICL on tabular data faced several fundamental obstacles [16, 27, 18, 52]. Chief among them is the highly inefficient tokenization of numerical tabular data into text, which quickly saturates the LLM's context window even for moderately sized tables. LLMs' results also vary based on the specific prompt format [55, 63] and are sensitive to the order of the given examples [43], whereas tabular data is inherently unordered. These limitations degrade performance, leading to prompt-tuning reliance and difficulty handling even moderately sized tables. Alternative ICL solutions for tabular data, such as TabPFN [47, 31], are architecturally designed for tabular data, enabling them to handle tables of practical size more effectively. This direction is gaining popularity but is still in early stages with relatively few tabular-based TFMs developed [32, 51]. Further investigation is needed into architecture design choices and training procedures that lead to strong downstream generalization, and in this work we make a major step in this direction.

We show that ICL retrieval combined with self-supervised learning (SSL) based on column masking [11, 28, 48] leads to a robust pre-training procedure for TFMs. We investigate the utility of real data, showing that using it for this pre-training procedure leads to faster convergence and improved downstream accuracy on unseen datasets compared to training exclusively on competing open-source synthetic data generators. Since existing TFMs are predominately trained with synthetic data, our findings suggest that further investigation should be conducted into the benefits of curating real datasets for pre-training. Our pre-training process produces a TFM capable of both classification and regression with leading accuracy on new, unseen datasets *without* any further training or tuning. We name this new model the Tabular Discriminative Pre-trained Transformer, or **TabDPT** for short.

Comprehensive evaluations on the OpenML-CC18 [5] and OpenML-CTR23 [17] benchmarks confirm the effectiveness of TabDPT. It consistently matches or surpasses the performance of specialized models that undergo extensive per-dataset hyperparameter optimization at a fraction of the deployment time and cost. Furthermore, we show strong results in the few-shot regime, where, with minimal semi-supervised modifications, TabDPT outperforms specialized baselines on 10-shot classification tasks, highlighting its versatility. Finally, we demonstrate that TabDPT scales predictably with both model size and quantity of real pre-training data (Figure 1), underscoring the viability of large-scale foundation models for tabular domains. We summarize our contributions as follows:

1. We develop a procedure for pre-training of TFMs based on ICL retrieval and SSL that leads to robust downstream generalization to unseen data without explicit fine-tuning.

2. We show that applying our pre-training procedure to real data leads to faster convergence and better downstream accuracy than using purely open-source synthetic data generators. We also demonstrate scaling with this procedure where more data and larger models continue to yield consistent gains akin to scaling laws in LLMs.

3. We release TabDPT – including open model weights, code, and pre-training datasets – to encourage further research and reproducibility. A lightweight inference interface is available here, and the full training code here.

Since its initial publication, TabDPT has also been compared externally against other tabular models – including other TFMs – on the popular TabArena [15] benchmark, available online here. The most recent results, as of January 14, 2026, demonstrate that TabDPT is the **premier open source TFM**:

- TabDPT has the highest ELO on all tasks among fully open source TFMs, and is third overall on the leaderboard (closely behind RealMLP [33] and Real TabPFN-2.5 [26]; the former is not a TFM and the latter has closed source training).

- TabDPT has the highest default performance on regression **of *any* model**, and is neck-and-neck with Real TabPFN-2.5 after tuning and ensembling.

## 2 Related Work

**Tabular Foundation Models** Although foundation models in other domains [11, 8, 12] have shown tremendous progress in recent years, foundation models for tabular data have lagged behind [59]. Several attempts have tried to bridge this gap. However, many of these methods require additional training when applied to a novel task [39, 60, 41], hindering widespread adoption in practice due to the high costs associated with fine-tuning. Meanwhile, ICL-based TFMs have started gaining traction. Among them, large language model (LLM) based approaches [18, 29, 57, 64] initially appear to be a natural fit. However, LLMs cannot easily handle numerical content of tables since their tokenization is not tailor-made for numerical data. As a result, LLM-based ICL methods suffer from high memory costs and low performance, as we discuss in Sections 3.1 and 4.3. Furthermore, natural language follows a sequential left-to-right structure, whereas tabular data is inherently order-invariant with respect to rows. Indeed, LLMs are sensitive to the order of examples in context [43].

On the other hand, tabular-specific ICL methods such as TabPFN [31] can naturally handle tabular data with numerical entries. However, they are completely reliant on synthetic data generators; ensuring that this mechanism captures the full diversity and nuances of real-world data is challenging, and making meaningful improvements to it is difficult. Notable concurrent work by Hollmann et al. [32] does not include open-sourced pre-training and synthetic generators, further complicating direct improvements to their model; another concurrent method requires a complex, three-stage procedure to learn from synthetic generators [51]. In this paper, we hypothesize that real tabular data contains much more information than existing heavily engineered synthetic tabular generators, thus allowing more straightforward improvements by scaling model and data size; this is supported by experiments in Section 4.5.

**Scaling Laws** Neural scaling laws have been studied extensively in various data modalities [8, 37]. In natural language processing (NLP), scaling laws were first identified in language models, where performance improves predictably with larger models, training corpora, and compute. Similar trends have been observed in computer vision [68]. Recently, Schambach et al. [54] demonstrated preliminary evidence of scaling laws for tabular data with very small-scale experiments. In this paper, we follow the developments from NLP, conducting thorough experiments to show that TabDPT follows scaling laws. Our novel analysis of scaling in the tabular domain paves the way for TFMs to scale and improve, much like foundation models in other domains.

**Tabular Self-supervised Learning** SSL has proven to be successful for text and images [11, 12], but has not achieved similar success on tabular data. Many tabular SSL methods cannot generalize beyond the dataset on which they were pre-trained [36, 67, 45, 56], raising the question of their potential to benefit from cross-task training. The answer is likely to be "yes", as recent work shows even tree-based methods benefit from hyperparameter tuning across tasks [33], and basic MLPs can be competitive in predictive tabular tasks when leveraging SSL [53]. Consequently, tabular SSL methods have begun to show generalization across tasks and competitive performance [69, 66]. However, they still require task specific fine-tuning and hyperparameter selection, which can be time- and resource-intensive. The only other tabular SSL method we are aware of that generalizes across tasks without per-task tuning is from Gardner et al. [18]. However, despite having 8 billion parameters (several orders of magnitude larger than TabDPT), its performance remains uncompetitive as its LLM-based design limits its context size to only 32 data points. To our knowledge, we are the first to demonstrate competitive performance and generalization of tabular SSL across tasks without task-specific training or hyperparameter tuning.

## 3 TabDPT Methodology

We now describe TabDPT, our approach for building a TFM, which employs (i) a row-based transformer encoder for in-context learning, (ii) self-supervised learning for *augmenting* the pre-training set, and (iii) retrieval-based context selection for both training and inference. These components are combined in a novel fashion to produce a single foundation model that generalizes to a diverse array of unseen classification and regression tasks without dataset-specific fine-tuning.

### 3.1 Tabular Transformer Encoder

We use a row-based transformer encoder similar to TabPFN v1 [31], where each table row serves as a "token." Specifically, for an input table with $N$ rows and $F$ features, we standardize its feature

dimension to $F_{\max}$ via padding ($F < F_{\max}$) or dimensionality reduction ($F > F_{\max}$), then embed each row into a $d$-dimensional vector. Rows attend to each other through stacked transformer layers.

A key motivation behind row-based encoding is memory and compute efficiency. In cell- or text-based tokenizations [22, 60, 32], each cell in an $N \times F$ table must be split into multiple tokens (e.g., subwords), resulting in $\mathcal{O}(N \times F \times \langle N_{\text{tok}} \rangle)$ tokens, where $\langle N_{\text{tok}} \rangle$ is the average number of tokens per cell. Even modest-sized tables can inflate the input sequence well beyond typical transformer context limits. By contrast, encoding *entire rows* as tokens reduces the sequence length to $N$, allowing us to process many more rows with significantly lower memory overhead. It also affords invariance to row ordering (no positional encoding), which matches the perceived structure of tabular data. While cell-based tokenization additionally introduces an appealing invariance to column ordering, we can achieve it approximately by permuting columns at training time; other works [32, 51] have also found that strict column order invariance can introduce training instability.

Finally, TabDPT uses a shared architecture for both regression and classification, which has been concurrently explored in TabPFN v2 [32]. In our case, this is realized through two separate MLP heads atop one single shared transformer: one head used for classification (supporting up to $C_{\max}$ classes) and another for regression. The shared backbone facilitates better parameter sharing across regression and classification tasks. Additional implementation details are provided in Appendix C.

## 3.2 Self-Supervised Learning on Tabular Data

Real-world tabular datasets are typically structured as $\mathcal{D} = \{X, y\}$, where $X \in \mathbb{R}^{N \times F}$ is the input table containing $N$ rows and $F$ features, and $y \in \mathbb{R}^{N \times 1}$ is the target (class index or regression value). In some instances $y$ can contain multiple targets but that is relatively rare. As the number of high quality publicly available tabular datasets is also relatively low, training TFMs in a supervised fashion to predict $y$ quickly saturates and leads to overfitting. To circumvent this, current methods predominantly leverage synthetic data that is continuously generated throughout pre-training [31, 7, 32, 51]. However, this approach has its own set of challenges where priors that generate synthetic data need to be extensively engineered to approximate the distribution of highly heterogeneous real-world data.

In this work we take a different approach and aim to maximize the value of real data in TFM pre-training. To this end, we leverage self-supervised learning (SSL) to extract maximal information from each table and regularize the model. Inspired by masked modeling in language [11] and vision [28], as well as some initial efforts in tabular domain [48, 18], we randomly designate one column as the "target" to be predicted from the others. Concretely, we randomly pick a task to be either regression or classification, then pick a column $c$ with sufficient unique values as the target. We remove $c$ from the table and standardize its values for regression or bin them into classes for classification. The model then has to predict this auxiliary target $y = c$ from the resulting table $X \backslash c$.

We also shuffle and drop other columns, forcing the model to learn from varying feature combinations. Without these augmentations, the number of prediction tasks would grow only linearly with the number of features. In contrast, our approach scales task count combinatorially, compelling the model to capture richer inter-feature relationships. This provides a stronger training signal and serves simultaneously as a regularizer. Pseudo-code for the SSL procedure is provided in Appendix D.

## 3.3 Retrieval-Based Pre-Training

In ICL, training data rows are passed as context to the transformer together with a target test row to generate a prediction. Although row-based embeddings allow for larger sample sizes, using full training tables as context still quickly becomes prohibitively large. Retrieval-based techniques, where only the top $K$ most similar training rows are selected as context, have been shown to mitigate this inherent

---

**Algorithm 1** One Training Step of TabDPT

Select $B$ random datasets $\{\mathcal{D}^{(i)}\}_{i=1}^{B}$
**for** *each dataset* $\mathcal{D}^{(i)}$ **do**
    Randomly set task as regression or classification
    Generate target $y^{(i)}$ from a random column $c^{(i)}$
    Sample $K$ "close" rows from $X^{(i)} \backslash c^{(i)}$
    Split rows into context $\{X_{\text{ctx}}^{(i)}, y_{\text{ctx}}^{(i)}\}$ and query $\{X_{\text{qy}}^{(i)}, y_{\text{qy}}^{(i)}\}$
    Shuffle and/or drop columns from $X_{\text{ctx}}^{(i)}$ and $X_{\text{qy}}^{(i)}$
Get transformer predictions $\{\hat{y}_{\text{qy}}^{(i)}\}_{i=1}^{B}$ (Equation 1)
Calculate loss and perform model update

---

limitation at *inference time* [58, 65], significantly improving the accuracy and scalability of ICL.

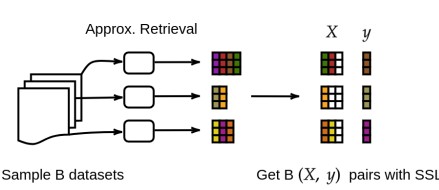
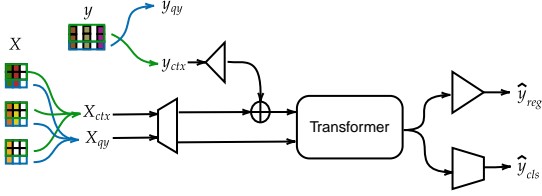

| (a) Selecting a training batch | (b) Overview of the architecture |

Figure 2: (a) We sample $B$ tables from different datasets to construct $X \in \mathbb{R}^{B \times N \times F_{\max}}$ and $y \in \mathbb{R}^{B \times N}$. (b) $X$ and $y$ are partitioned into context $\{X_{\text{ctx}}, y_{\text{ctx}}\}$ and query $X_{\text{qy}}$ inputs and passed through embedding functions (indicated by rectangle/triangle). Embeddings of $X_{\text{ctx}}$ and $y_{\text{ctx}}$ are summed together, concatenated with context embedding of $X_{\text{qy}}$, and passed through a transformer encoder to get classification $\hat{y}_{\text{cls}}$ or regression $\hat{y}_{\text{reg}}$ prediction for the query. Loss between this prediction and query targets $y_{\text{qy}}$ is used to update the model.

We propose to take this one step further and align training with inference by also leveraging retrieval during training batch construction. Formally, after obtaining $y = c$ and $X \setminus c$ through SSL, we sample a set of $K$ rows from $X \setminus c$ that are close to each other in the feature space. These rows are partitioned into two groups uniformly at random: "context" $\{X_{\text{ctx}}, y_{\text{ctx}}\}$ and "query" $\{X_{\text{qy}}, y_{\text{qy}}\}$. The context $\{X_{\text{ctx}}, y_{\text{ctx}}\}$, together with the query features $X_{\text{qy}}$, is fed into the model to predict query targets $y_{\text{qy}}$. To form model inputs, we pass the context and query through the appropriate row embedding functions ($\phi_x$ or $\phi_y$), then sum the embeddings of $X_{\text{ctx}}$ and $y_{\text{ctx}}$, and concatenate with $X_{\text{qy}}$:

$$\hat{y}_{\text{qy}} = \textbf{Transformer}\left[\phi_x(X_{\text{ctx}}) \oplus \phi_y(y_{\text{ctx}}), \phi_x(X_{\text{qy}})\right]. \tag{1}$$

Here, $\oplus$ is element-wise addition, and $\hat{y}_{\text{qy}}$ emerges from the classification or regression head depending on the target task. The notation $[\cdot, \cdot]$ indicates the cross-attention split: query points attend to all context points, while context points attend to each other but not to queries – that is, only context serves as keys in the attention mechanism.

Using "similar" rows as context during training and limiting context size naturally aligns it with inference, improving model generalization. It also maintains efficient batch sizes, as context size no longer scales with table size, while still exposing the model to relevant context information. We consistently observe that training with this procedure speeds up convergence and leads to better downstream accuracy. An illustration of our model architecture is shown in Figure 2, with a full training step described in Algorithm 1.

### 3.4 Inference on New Data

At inference, given a new dataset, we follow the same retrieval protocol. For each test query row $x_{\text{qy}}$ we retrieve the top $K$ closest rows from the training set to get context $\{X_{\text{ctx}}, y_{\text{ctx}}\}$. Context and query inputs are then embedded and passed through the transformer encoder (see Equation 1) to get classification $\hat{y}_{\text{cls}}$ or regression $\hat{y}_{\text{reg}}$ predictions for the query row, depending on the target task. Note that other than context retrieval, no dataset-specific tuning is done and we only run forward passes through the model. Retrieval can add additional latency, however, modern nearest neighbor libraries such as FAISS [13] deliver millisecond responses and scale to billion-row indices. Our backbone imposes a pre-defined maximum number of classes $C_{\max}$ and features $F_{\max}$; we now discuss how to overcome these limitations with inference-time techniques.

**Classes** If a dataset contains $C > C_{\max}$ classes, we cannot perform classification in a single forward pass. Naive binary one-versus-all classification would require $C$ forward passes that can significantly impact inference speed as some datasets have hundreds of classes. A more computationally efficient approach is to represent $C$ in base $C_{\max}$ and perform classification on each base-$C_{\max}$ digit as a separate, well-defined prediction task. This approach reduces the required number of forward passes to $\lceil \log_{C_{\max}}(C) \rceil$ and is fully compatible with the TFM setting.

**Features** When the number of features in a table exceeds $F_{\max}$, we can reduce the dimensionality of the table using Principal Component Analysis (PCA) to $F_{\max}$, effectively compressing the features to fit the model requirement, while preserving the most salient information.

# 4 Experiments

In this section, we evaluate TabDPT against leading baselines on standard benchmarks for TFMs, provide a detailed analysis of runtime, and ablate key components.

## 4.1 Data

**Training Data** Our training data was collected from OpenML [61] and consists of a wide range of public tabular datasets across diverse domains, all available under the CC-BY licence. To find appropriate datasets, we considered those specified in the Grinsztajn et al. [25], TabZilla [46], and AMLB [19] benchmarks, as well as additional datasets found individually. The full set of pre-training data contains 123 datasets, with a total of 32M rows and 2B cells (individual values within each table) from a diverse set of domains such as biology, finance, industrial applications, and medicine. The scale of this data is comparable to related work such as Tabula-8B [18] that fine-tuned the LLaMA 3-8B [24] language model for the tabular domain using real-world data. We conjecture that the diversity of domains present in our pre-training data can provide a salient signal and improve downstream generalization. Further details, including the complete list of training datasets and breakdown by size and domain are provided in Appendix B.

**Evaluation Data** For evaluation, we consider two commonly used public benchmarks containing a total of 107 datasets: CC18 [5] for classification tasks and CTR23 [17] for regression tasks. CC18 is a suite of 72 classification datasets originally sourced from OpenML. These datasets contain between 500 and 100,000 instances, fewer than 5,000 features, and originate from diverse domains such as finance, biology, games, banking, industrial applications, or natural signals such as vision or sound. Datasets were selected according to curation criteria that included removing synthetic data, requiring source information, and removing datasets where a simple algorithm achieves $100\%$ accuracy. CC18 is a common benchmark for evaluating tabular learning on classification tasks [2, 31, 46]. CTR23 is a benchmark suite of 35 datasets also curated from OpenML. It follows most of the design choices of CC18 but contains only regression tasks. In particular, it uses the same restrictions on the number of samples and features as CC18, but replaces the accuracy restriction with a requirement that a linear model must not achieve $R^2 = 1$.

## 4.2 Baselines

We compare our method against leading baselines that are tuned for each dataset, including tree-based methods such as XGBoost [9], LightGBM [38], and CatBoost [50], and deep learning methods such as TabR [22], TabM [23] and MLP-PLR [21], as well as MLP. For XGBoost, CatBoost, and LightGBM, we use results reported in the TabZilla benchmark [46]. Some datasets are missing results, so we conduct hyperparameter optimization and train models following the TabZilla protocol using the code repository from [22].[1] For TabR, TabM, MLP-PLR, and MLP, we use the same code repository with the predefined search space and 30 search rounds for both CC18 and CTR23. We choose the best hyperparameters for each dataset fold individually based on the validation performance.

We also compare to ICL baselines including the LLM-based Tabula-8B [18], and tabular-specific foundation models TabPFN v2 [32], and TabPFN (kNN) [58], which retrieves neighbours of each query at inference time. Note that using retrieval with a cell-based method like TabPFN v2 adds prohibitive computational overhead. We run all methods on at least two different splits of the data and report $95\%$ confidence intervals using bootstrapping [1]. For TabDPT, we use the 78M-parameter variant, with 16 transformer layers pre-trained for 600K steps. All training and inference is done on Nvidia A100 GPUs with 40 GB of memory. Further training details are provided in Appendix C

## 4.3 Results on CC18 and CTR23

Our main results comparing models on the evaluation data are shown in Table 1. TabDPT shows the best overall performance across all models. It is competitive with TabPFN v2 across metrics, and significantly outperforms it on accuracy on CC18. TabDPT also significantly outperforms both deep learning and tree-based methods that are trained for each dataset. These results indicate that real data can be effectively utilized with SSL to train robust TFMs with leading performance. We

---

[1] `https://github.com/yandex-research/tabular-dl-tabr`

| Algorithm | CC18 | | CTR23 | |
| --- | --- | --- | --- | --- |
| | **AUC** | **Accuracy** | **Correlation** | $R^2$ |
| TabDPT | **0.933** ± [0.929, 0.937] | **0.884** ± [0.882, 0.887] | **0.837** ± [0.826, 0.848] | **0.742** ± [0.731, 0.754] |
| TabPFN v2 | 0.932 ± [0.928, 0.936] | 0.872 ± [0.869, 0.875] | 0.835 ± [0.825, 0.845] | 0.740 ± [0.729, 0.751] |
| TabPFN (kNN) | 0.918 ± [0.915, 0.921] | 0.850 ± [0.847, 0.853] | N/A | N/A |
| TabPFN | 0.898 ± [0.895, 0.901] | 0.812 ± [0.810, 0.814] | N/A | N/A |
| TabDPT (no val.) | 0.928 ± [0.924, 0.932] | 0.874 ± [0.871, 0.877] | 0.835 ± [0.826, 0.844] | 0.739 ± [0.729, 0.749] |
| TabM | 0.917 ± [0.913, 0.921] | 0.878 ± [0.876, 0.881] | 0.824 ± [0.815, 0.834] | 0.732 ± [0.722, 0.742] |
| TabR | 0.925 ± [0.922, 0.928] | 0.874 ± [0.871, 0.877] | 0.828 ± [0.814, 0.842] | 0.714 ± [0.693, 0.735] |
| MLP-PLR | 0.912 ± [0.905, 0.919] | 0.869 ± [0.865, 0.872] | 0.829 ± [0.821, 0.838] | 0.716 ± [0.699, 0.732] |
| MLP | 0.872 ± [0.867, 0.876] | 0.809 ± [0.805, 0.813] | N/A | N/A |
| XGBoost | 0.926 ± [0.922, 0.929] | 0.869 ± [0.866, 0.872] | 0.827 ± [0.818, 0.837] | 0.711 ± [0.698, 0.725] |
| LightGBM | 0.924 ± [0.920, 0.927] | 0.862 ± [0.859, 0.866] | 0.825 ± [0.816, 0.834] | 0.713 ± [0.697, 0.729] |
| CatBoost | 0.926 ± [0.923, 0.930] | 0.864 ± [0.860, 0.867] | 0.822 ± [0.808, 0.837] | 0.703 ± [0.683, 0.723] |

Table 1: Main results comparing models on evaluation data. We report the average performance on four metrics and their 95% confidence intervals. The best algorithm for each metric is bolded. The best model that does not append the validation set to the training set is underlined. Tabula-8B [18] only reports results on a subset of datasets in CC18 so we conduct pairwise comparison against it on reported datasets in Figure 3a.

provide a breakdown of results in Appendix F.1, examining each algorithm's performance under varying dataset sizes, numbers of features, categorical fraction, and percent missing. Results indicate that TabDPT is robust to dataset variations in all of these categories. For Table 1, we use TabDPT with 2,048 context size and 8 ensemble members, where randomness for a given ensemble member emerges by permuting the features and classes (if applicable).

Our foundation models require negligible hyperparameter tuning. Rather than reserving a validation set, we merge it with the training set. In contrast, several baselines perform hyperparameter tuning on a separate validation set (instead of cross-validation with a final refit on the combined train and validation set). To benchmark under the same constraint, we report a variant in which TabDPT receives only the training set, denoted by (no val.). This setting should be compared to classical methods; the default TabDPT setting is comparable to other foundation models that also access the validation set. Since TabDPT (no val.) never uses validation data, it is disadvantaged compared to methods that tune on validation data. Even so, TabDPT (no val.) remains competitive and is only outperformed by TabM [23] (using 30 hyperparameter tuning rounds) on CC18 accuracy.

**Win-Rate Comparison** To get a more direct view of how various methods perform against each other, we compute pairwise win-rate statistics. We assign a "win" to the method if it achieves a higher accuracy score on CC18 or $R^2$ score on CTR23. This also allows us to compare against Tabula-8B on CC18 as it only reports results on 61 of 72 datasets. Figure 3a shows the win-rate matrix for all methods, painting a similar picture to Table 1: TabDPT performs best overall, followed by the strong tabular-based foundation model TabPFN v2. Tabula-8B with 32 shots – the leading LLM-based tabular foundation model – is not competitive with the other techniques across our benchmarks, indicating that current LLMs are not well adapted to the tabular domain and further techniques are needed. We extend the pairwise comparison between methods in Appendix E using both Elo [14] and Glicko2 [20] metrics, drawing similar conclusions to above.

## 4.4 Ablation Study

In this section, we discuss the ablation of key components in our training and inference strategies, with results visualized in Figure 4.

**Training Ablation** First, we assess the importance of SSL during training. To ablate SSL, we only use the original target for each table during training and observe a large loss in performance, as shown under "Supervised Target (Tr)" in Figure 4a. The impact on training is further illustrated by the "Real-No SSL" curve in Figure 4b. Training without SSL starts to overfit after around 50 epochs whereas with SSL model continues to improve even after 500 epochs ("Real - SSL" curve). These

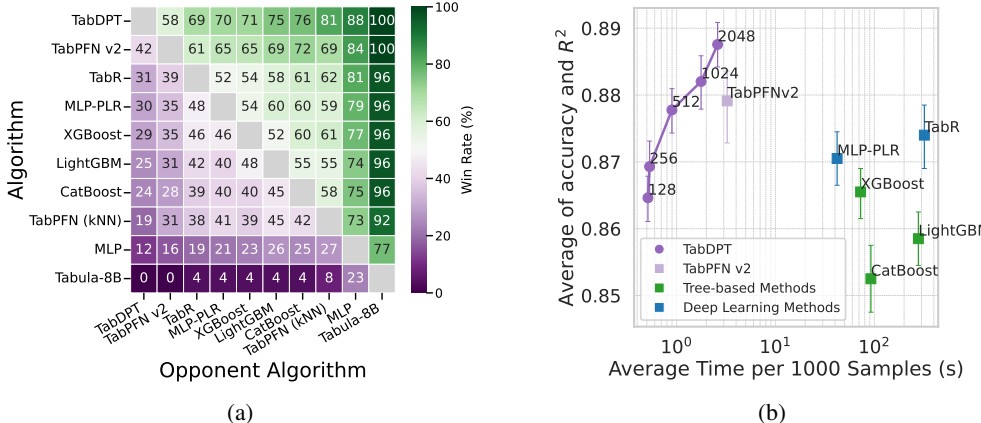

(a)                     (b)

Figure 3: (a) Pairwise win-rate comparison. A win is counted for the method that achieves the higher classification/regression accuracy/$R^2$ on a given dataset. (b) Inference runtime vs performance. Tab-DPT models are ordered by context size. Non-TFM baseline runtimes are the total of hyperparameter optimization and inference.

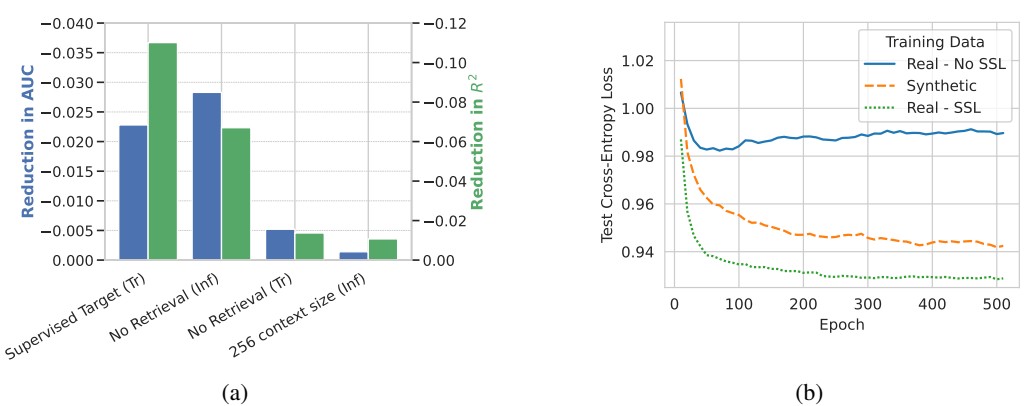

(a)                     (b)

Figure 4: (a) Ablation of key components in training (Tr) and inference (Inf). A higher blue bar and a higher green bar indicate greater reduction in AUC and $R^2$ respectively. (b) Test loss curves on CC18 when training with and without SSL on real data as well as synthetic data only. We note that "Synthetic" and "Real - SSL" both saturate at 500 epochs, which we confirmed by training for 1024 epochs.

results demonstrate the critical role of SSL when training TFMs with real data where only one target is typically available per dataset.

Second, we assess the benefit of using retrieval during training to form the context versus random subsampling, the results are shown under "No Retrieval (Tr)" in Figure 4a. We see that removing training retrieval leads to a consistent drop in both classification and regression test accuracy. Although smaller in magnitude than drop from removing SSL, these results confirm that aligning training and inference procedures is beneficial.

Finally, we benchmark the impact of training on real ("Real - SSL" curve) vs. synthetic data ("Synthetic" curve) in Figure 4b. We use the TabPFN [31] synthetic data generator for this experiment. We see that using real data with SSL consistently outperforms training on synthetic data across all epochs, achieving lower test loss under the same compute budget. This further highlights the effectiveness of real data when paired with SSL on our TFM architecture.

**Inference Ablation** Similarly to [58], we find that using subsampling instead of retrieval during inference significantly decreases performance as indicated by "No Retrieval (Inf)" in Figure 4a. Using a smaller context of 256 size also decreases performance as expected, although it does not decrease nearly as much as the other important components discussed above.

| Method | cmc | karhunen | optdigit | diabetes | semeion | pixel | dna | Avg. |
|---|---|---|---|---|---|---|---|---|
| TabDPT (semi) | **44.24** | **92.08** | **94.31** | 72.01 | **84.89** | **93.58** | 61.85 | **77.56** |
| STUNT [48] | 42.01 | 86.95 | 89.91 | **72.82** | 74.74 | 89.90 | 80.96 | 76.76 |
| CACTUs [35] | 42.14 | 85.48 | 87.92 | 70.75 | 68.22 | 87.21 | **84.40** | 75.16 |
| VIME + LR[67] | 37.92 | 86.63 | 89.63 | 66.56 | 77.66 | 88.71 | 74.73 | 74.55 |
| kNN (STUNT) [48] | 41.07 | 85.63 | 87.44 | 71.32 | 74.64 | 87.52 | 71.15 | 74.11 |
| ICT [62] | 38.00 | 88.25 | 90.84 | 67.63 | 74.67 | 89.13 | 69.55 | 74.01 |

Table 2: Few-shot accuracy on seven CC18 datasets. Only 10 labeled examples are available in each class of the training set, the rest are unlabeled.

## 4.5 Scaling Laws

Although preliminary results on tabular scaling have been reported [54], this work provides the first analysis of scaling laws for TFMs that are not restricted to any particular domain. We focus on measuring scaling with pre-training on real data only, and evaluate performance as a function of training data amount and model size, systematically varying both. Model size is varied by changing both the number of layers and their dimensions. Our models range from 33K to 78M parameters, trained on data subsets spanning from 52M cells (104K rows) to 2B cells (32M rows). Following Hoffmann et al. [30], we adopt the joint power-law model:

$$\hat{\ell}(P, D) = AP^{-\alpha} + BD^{-\beta} + E \tag{2}$$

where $\hat{\ell}$ represents the estimated target metric, $P$ and $D$ denote the number of parameters and total cells in the training set, and $A, B, \alpha, \beta, E$ are the scaling parameters. Despite using a row-based encoding, we measure data size by cell count, as not all rows contribute equally to the model's learning, particularly in the encoder layer that computes the embeddings. Applying the improved methodology of Besiroglu et al. [4], we estimate the scaling exponents as $\alpha = 0.42$ and $\beta = 0.39$, indicating that improvements can occur in both dimensions.

In Figure 1, we illustrate the scaling behaviour of our models along with the power-law fit. Since we train on equal proportions of classification and regression tasks, the loss on the $y$-axis represents the mean of the cross-entropy loss for classification and $1 - \rho$ for regression, where $\rho$ is the correlation between prediction and target, equivalent to the MSE for normalized vectors. Visualization is done on a log scale for the excess loss $\hat{\ell}(P, D) - E$ instead of the raw loss, debiasing the estimate by $E$. We observe consistent improvements when both data and model size increase indicating that information contained in real world tabular datasets can be effectively mined with SSL to pre-train robust TFMs.

## 4.6 Few-Shot Learning

We next assess the performance of TabDPT on few-shot learning tasks. We adopt the protocol from STUNT [48] with the 10-shot set-up where only 10 labeled rows are provided for each class in each training table, the rest are masked and the goal is to leverage the labeled + unlabeled data to accurately predict the test set. This simulates real-world settings where only small subsets of data can have labels. We compare against baseline results from [48] including STUNT [48], CACTUs [35], VIME+LR [67], and ICT [62] available in their paper. All methods are evaluated on seven datasets from CC18 using the accuracy metric.

TabDPT typically leverages a much larger context than 10 instances during both training and inference. To adapt it to the few-shot setting, we increase the context size by first predicting class probabilities for the unlabeled training set using the 10 labeled examples as context. Then, we take the top-1000 points where predicted probability is highest and use them and their predicted labels – along with the original 10 shots – as context. This results in TabDPT (semi), a semi-supervised version of our TFM that leverages pseudo-labels. This method outperforms STUNT, a leading few-shot method, on 5 of the 7 datasets and on average accuracy (over 50 seeds). Furthermore, it requires only forward passes to generalize to new tasks once we have a pre-trained model, while STUNT trains a new model for each task. This experiment demonstrates the potential of TabDPT as a TFM: it can rapidly adapt to new tabular settings without any additional weight updates.

Concurrently, Li et al. [40] found that TFMs performed strongly in the semi-supervised setting using a method distinct to ours; combining their approach with TabDPT is a promising avenue for future work.

## 4.7 Inference Speed

In this section, we analyze the inference runtime of TabDPT against baselines on new datasets. For TFMs we measure the cost of computing the context and making forward passes through the models. For deep learning and tree-based methods that are dataset specific we measure the total time to train the model, including hyperparameter search, and to run inference. We repeat each experiment across dataset folds and measure the average times to process 1,000 rows (computed overall on $\approx 2M$ rows). For TabDPT we report runtimes with different context sizes from 128 to 2048 and the corresponding impact on accuracy. Results are shown in Figure 3b, we see that even our largest model with context size 2048 is at least one order of magnitude faster than dataset-specific tree and deep learning baselines. TabDPT runtime is also comparable to TabPFN v2 while achieving higher accuracy. The cost of pre-training TFMs is not included in this comparison. However, analogous to LLMs, we stipulate that it is a one-time cost that is offset when model is applied across many datasets and use cases.

On the other hand, when training cost is not part of the budget (e.g., fixed models on fixed datasets), classical tree-based pipelines can deliver substantially faster inference (e.g., $\sim$0.02–0.10s per 1,000 rows), often about an order of magnitude faster than TabDPT for inference alone.

## 5  Conclusion and Future Work

We introduce TabDPT, an open tabular foundation model with a demonstrated ability to generalize on a range of unseen tabular datasets without additional tuning or adaptation. Our training approach provides an effective way to leverage real data with SSL to build robust TFMs. Models pre-trained with our procedure exhibit scaling laws with consistent improvements from both data and model size, analogous to foundation models in other domains. Given the practical ease of use and broad applicability of foundation models, we believe that these contributions will advance their adoption as an alternative to individually trained models in tabular domains.

While TabDPT demonstrates strong performance, opportunities remain to further enhance TFMs – including TabDPT – and address current limitations. (i) Preliminary experiments using feature name embeddings led to overfitting. Expanding training data with more diverse tables containing free-form text may mitigate this and improve textual integration. (ii) TabDPT is designed for rectangular datasets with i.i.d. rows and does not explicitly model temporal dependencies, distribution shifts, or hierarchical structures. Methodological improvements could help overcome these constraints; e.g., ideas similar to Hoo et al. [34] can help tackle temporal dependencies. (iii) TabDPT could be enhanced with techniques from works such as Ma et al. [44] and van Breugel et al. [60] to complement its discriminative capabilities with generative modeling and density estimation.

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

# A    Bitter Lessons

Throughout the development of TabDPT we tried many variations to improve performance and/or scalability. Some were successful while others did not lead to an improvement. We list some of these variations here to facilitate future research on TabDPT and related architectures. Overall, our findings align with *The Bitter Lesson*[2] where efficient use of computation and access to high-quality data are much more important for driving performance than architectural manipulations.

- Different pre-processing techniques that were more robust to outliers, or variants of soft clipping, resulted in no improvement. More advanced methods, such as Robust Scaler and Power Transform, only ended up slowing the training process.

- Class embeddings (either through a separate network or by using class "tokens" in the transformer layer) and computing various similarity metrics between query and class embeddings in a proto-network manner, with the aim of adapting to any number of classes, hurt the performance, especially on real data.

- Different embeddings for $y_{\text{ctx}}$, including a dense layer for regression and a dictionary of $C_{\max} \times d$ embeddings, with the rationale of informing the model about the task, did not lead to performance improvements in large models with sufficient data.

- Specialized tokens for NaN encoding did not improve performance compared to replacing NaNs with mean values (which are zero after preprocessing). Additionally, appending binary features to differentiate actual zeros from NaNs (indicating that the cell was replaced), effectively doubling the number of features, also failed to improve performance.

- Architectures encoding cells as "tokens", with vertical and horizontal attention, similar to spatial and temporal attention in videos, proved more memory intensive. While equivariance to feature order is desirable, processing tensors of size $(B, N, f, d)$ – where $B$ is batch size, $N$ is the number of rows, $f$ the number of features, and $d$ the embedding dimension – uses much more memory. The simpler architecture with tensors of size $(B, N, d)$ permits a higher embedding dimension $d$. While Hollmann et al. [32] is able to make this architecture work, we suspect that the differences between synthetic and real data are enough to change which architectures are performant.

We would also like to emphasize that these *Bitter Lessons* reflect our own experience in training an ICL-based TFM with real data. Your mileage may vary.

# B    Training Datasets

Figure B.1 provides a summary of the sizes and domains of the training datasets, and Table B.1 provides a full list of the datasets. Note that 93 datasets have classification targets, 29 datasets have regression targets, and 1 does not have a default target defined. However, we generate both classification and regression targets for each dataset by applying the SSL procedure described in Section 3.

Table B.1: Details for all training datasets: OpenML Dataset ID, name, dimensions (rows, features, cells), percent of missing cells, target type (classification/regression), domain.

| OpenML Dataset ID | Name | # rows | # feat. | # cells | % miss. | Target type | Domain |
|---|---|---|---|---|---|---|---|
| 24 | mushroom | 8124 | 22 | 187K | 1.4 | Class. | Biology/ecology |
| 30 | page-blocks | 5473 | 10 | 60K | 0.0 | Class. | Vision/audio/text features |
| 184 | kropt | 28056 | 6 | 196K | 0.0 | Class. | Deterministic and simulated |
| 273 | IMDB.drama | 120919 | 1001 | 121M | 0.0 | Class. | Other or not provided |
| 312 | scene | 2407 | 299 | 722K | 0.0 | Class. | Vision/audio/text features |
| 375 | JapaneseVowels | 9961 | 14 | 149K | 0.0 | Class. | Vision/audio/text features |
| 382 | ipums_la_97-small | 7019 | 60 | 428K | 11.4 | Class. | Financial/demographic |
| 389 | fbis.wc | 2463 | 2000 | 4.9M | 0.0 | Class. | Vision/audio/text features |
| 396 | la1s.wc | 3204 | 13195 | 42M | 0.0 | Class. | Vision/audio/text features |
| 802 | pbcseq | 1945 | 18 | 37K | 3.2 | Class. | Medical/human sensor |
| 816 | puma8NH | 8192 | 8 | 74K | 0.0 | Class. | Deterministic and simulated |
| 821 | house_16H | 22784 | 16 | 387K | 0.0 | Class. | Financial/demographic |

---

[2] http://www.incompleteideas.net/IncIdeas/BitterLesson.html

| OpenML Dataset ID | Name | # rows | # feat. | # cells | % miss. | Target type | Domain |
|---|---|---|---|---|---|---|---|
| 843 | house_8L | 22784 | 8 | 205K | 0.0 | Class. | Financial/demographic |
| 846 | elevators | 16599 | 18 | 315K | 0.0 | Class. | Other or not provided |
| 871 | pollen | 3848 | 5 | 23K | 0.0 | Class. | Biology/ecology |
| 930 | colleges_usnews | 1302 | 33 | 44K | 18.2 | Class. | Other or not provided |
| 966 | analcatdata_halloffame | 1340 | 16 | 23K | 0.1 | Class. | Other or not provided |
| 981 | kdd_internet_usage | 10108 | 68 | 697K | 0.4 | Class. | Financial/demographic |
| 1002 | ipums_la_98-small | 7485 | 55 | 419K | 7.9 | Class. | Financial/demographic |
| 1018 | ipums_la_99-small | 8844 | 56 | 504K | 7.0 | Class. | Financial/demographic |
| 1036 | sylva_agnostic | 14395 | 216 | 3.1M | 0.0 | Class. | Biology/ecology |
| 1037 | ada_prior | 4562 | 14 | 68K | 0.1 | Class. | Financial/demographic |
| 1043 | ada_agnostic | 4562 | 48 | 224K | 0.0 | Class. | Financial/demographic |
| 1044 | eye_movements | 10936 | 27 | 306K | 0.0 | Class. | Medical/human sensor |
| 1111 | KDDCup09_appetency | 50000 | 230 | 12M | 61.9 | Class. | Human behaviour |
| 1112 | KDDCup09_churn | 50000 | 230 | 12M | 61.9 | Class. | Industrial/operational |
| 1116 | musk | 6598 | 167 | 1.1M | 0.0 | Class. | Other science |
| 1118 | chess | 28056 | 6 | 196K | 0.0 | Class. | Deterministic and simulated |
| 1120 | MagicTelescope | 19020 | 10 | 209K | 0.0 | Class. | Physics/astronomy |
| 1130 | OVA_Lung | 1545 | 10935 | 17M | 0.0 | Class. | Biology/ecology |
| 1142 | OVA_Endometrium | 1545 | 10935 | 17M | 0.0 | Class. | Biology/ecology |
| 1169 | airlines | 539383 | 7 | 4.3M | 0.0 | Class. | Industrial/operational |
| 1444 | PizzaCutter3 | 1043 | 37 | 40K | 0.0 | Class. | Other or not provided |
| 1453 | PieChart3 | 1077 | 37 | 41K | 0.0 | Class. | Other or not provided |
| 1457 | amazon-commerce-reviews | 1500 | 10000 | 15M | 0.0 | Class. | Vision/audio/text features |
| 1459 | artificial-characters | 10218 | 7 | 82K | 0.0 | Class. | Deterministic and simulated |
| 1466 | cardiotocography | 2126 | 35 | 77K | 0.0 | Class. | Medical/human sensor |
| 1471 | eeg-eye-state | 14980 | 14 | 225K | 0.0 | Class. | Medical/human sensor |
| 1476 | gas-drift | 13910 | 128 | 1.8M | 0.0 | Class. | Other science |
| 1477 | gas-drift-different-concentrations | 13910 | 129 | 1.8M | 0.0 | Class. | Other science |
| 1479 | hill-valley | 1212 | 100 | 122K | 0.0 | Class. | Deterministic and simulated |
| 1481 | kr-vs-k | 28056 | 6 | 196K | 0.0 | Class. | Deterministic and simulated |
| 1483 | ldpa | 164860 | 7 | 1.3M | 0.0 | Class. | Medical/human sensor |
| 1493 | one-hundred-plants-texture | 1599 | 64 | 104K | 0.0 | Class. | Biology/ecology |
| 1503 | spoken-arabic-digit | 263256 | 14 | 3.9M | 0.0 | Class. | Vision/audio/text features |
| 1507 | twonorm | 7400 | 20 | 155K | 0.0 | Class. | Deterministic and simulated |
| 1509 | walking-activity | 149332 | 4 | 747K | 0.0 | Class. | Medical/human sensor |
| 1567 | poker-hand | 1025009 | 10 | 11M | 0.0 | Class. | Deterministic and simulated |
| 1568 | nursery | 12958 | 8 | 117K | 0.0 | Class. | Financial/demographic |
| 1596 | covertype | 581012 | 54 | 32M | 0.0 | Class. | Biology/ecology |
| 3050 | QSAR-TID-11 | 5742 | 1024 | 5.9M | 0.0 | Reg. | Medical/human sensor |
| 3277 | QSAR-TID-10980 | 5766 | 1024 | 5.9M | 0.0 | Reg. | Medical/human sensor |
| 4135 | Amazon_employee_access | 32769 | 9 | 328K | 0.0 | Class. | Industrial/operational |
| 4535 | Census-Income | 299285 | 42 | 13M | 0.0 | None | Financial/demographic |
| 4549 | Buzzinsocialmedia_Twitter | 583250 | 77 | 45M | 0.0 | Reg. | Human behaviour |
| 23380 | cjs | 2796 | 33 | 95K | 73.8 | Class. | Biology/ecology |
| 23512 | higgs | 98050 | 28 | 2.8M | 0.0 | Class. | Physics/astronomy |
| 40536 | SpeedDating | 8378 | 120 | 1.0M | 1.8 | Class. | Human behaviour |
| 40646 | GAMETES_Epistasis_2-Way_20atts_0.1H_EDM-1_1 | 1600 | 20 | 34K | 0.0 | Class. | Biology/ecology |
| 40679 | magic | 19020 | 10 | 209K | 0.0 | Class. | Physics/astronomy |
| 40680 | mofn-3-7-10 | 1324 | 10 | 15K | 0.0 | Class. | Other or not provided |
| 40685 | shuttle | 58000 | 9 | 580K | 0.0 | Class. | Physics/astronomy |
| 40706 | parity5_plus_5 | 1124 | 10 | 12K | 0.0 | Class. | Deterministic and simulated |
| 40733 | yeast | 1269 | 8 | 11K | 0.0 | Class. | Biology/ecology |
| 40900 | Satellite | 5100 | 36 | 189K | 0.0 | Class. | Physics/astronomy |
| 41138 | APSFailure | 76000 | 170 | 13M | 8.3 | Class. | Industrial/operational |
| 41142 | christine | 5418 | 1636 | 8.9M | 0.0 | Class. | Other or not provided |
| 41143 | jasmine | 2984 | 144 | 433K | 0.0 | Class. | Other or not provided |
| 41144 | madeline | 3140 | 259 | 816K | 0.0 | Class. | Other or not provided |
| 41145 | philippine | 5832 | 308 | 1.8M | 0.0 | Class. | Other or not provided |
| 41146 | sylvine | 5124 | 20 | 108K | 0.0 | Class. | Other or not provided |
| 41147 | albert | 425240 | 78 | 34M | 8.2 | Class. | Other or not provided |
| 41150 | MiniBooNE | 130064 | 50 | 6.6M | 0.0 | Class. | Physics/astronomy |
| 41156 | ada | 4147 | 48 | 203K | 0.0 | Class. | Other or not provided |
| 41159 | guillermo | 20000 | 4296 | 86M | 0.0 | Class. | Other or not provided |
| 41161 | riccardo | 20000 | 4296 | 86M | 0.0 | Class. | Other or not provided |
| 41162 | kick | 72983 | 32 | 2.4M | 6.4 | Class. | Industrial/operational |
| 41163 | dilbert | 10000 | 2000 | 20M | 0.0 | Class. | Other or not provided |
| 41164 | fabert | 8237 | 800 | 6.6M | 0.0 | Class. | Other or not provided |
| 41165 | robert | 10000 | 7200 | 72M | 0.0 | Class. | Other or not provided |
| 41166 | volkert | 58310 | 180 | 11M | 0.0 | Class. | Other or not provided |
| 41167 | dionis | 416188 | 60 | 25M | 0.0 | Class. | Other or not provided |
| 41168 | jannis | 83733 | 54 | 4.6M | 0.0 | Class. | Other or not provided |
| 41169 | helena | 65196 | 27 | 1.8M | 0.0 | Class. | Other or not provided |
| 41434 | Click_prediction_small | 39948 | 11 | 479K | 0.0 | Class. | Human behaviour |
| 41540 | black_friday | 166821 | 9 | 1.7M | 0.0 | Reg. | Human behaviour |
| 41980 | SAT11-HAND-runtime-Reg. | 4440 | 116 | 519K | 5.3 | Reg. | Computing |

| OpenML Dataset ID | Name | # rows | # feat. | # cells | % miss. | Target type | Domain |
|---|---|---|---|---|---|---|---|
| 42563 | house_prices_nominal | 1460 | 79 | 117K | 6.0 | Reg. | Financial/demographic |
| 42572 | Santander_transaction_value | 4459 | 4991 | 22M | 0.0 | Reg. | Human behaviour |
| 42705 | Yolanda | 400000 | 100 | 40M | 0.0 | Reg. | Other or not provided |
| 42724 | OnlineNewsPopularity | 39644 | 59 | 2.4M | 0.0 | Reg. | Human behaviour |
| 42727 | colleges | 7063 | 44 | 318K | 33.5 | Reg. | Other or not provided |
| 42728 | Airlines_DepDelay_10M | 10000000 | 9 | 100M | 0.0 | Reg. | Industrial/operational |
| 42730 | us_crime | 1994 | 126 | 253K | 15.6 | Reg. | Financial/demographic |
| 42732 | sf-police-incidents | 2215023 | 8 | 20M | 0.0 | Class. | Human behaviour |
| 42734 | okcupid-stem | 50789 | 19 | 1.0M | 16.0 | Class. | Human behaviour |
| 42742 | porto-seguro | 595212 | 57 | 35M | 2.5 | Class. | Human behaviour |
| 42746 | KDDCup99 | 4898431 | 41 | 206M | 0.0 | Class. | Computing |
| 43071 | MIP-2016-Reg. | 1090 | 144 | 158K | 0.0 | Reg. | Computing |
| 43072 | KDDCup09-Upselling | 50000 | 14891 | 745M | 2.6 | Class. | Human behaviour |
| 44055 | analcatdata_supreme | 4052 | 7 | 32K | 0.0 | Reg. | Other or not provided |
| 44056 | visualizing_soil | 8641 | 4 | 43K | 0.0 | Reg. | Biology/ecology |
| 44061 | Mercedes_Benz_Greener_Manufacturing | 4209 | 359 | 1.5M | 0.0 | Reg. | Industrial/operational |
| 44063 | Bike_Sharing_Demand | 17379 | 11 | 209K | 0.0 | Reg. | Human behaviour |
| 44065 | nyc-taxi-green-dec-2016 | 581835 | 16 | 9.9M | 0.0 | Reg. | Human behaviour |
| 44068 | particulate-matter-ukair-2017 | 394299 | 6 | 2.8M | 0.0 | Reg. | Other or not provided |
| 44069 | SGEMM_GPU_kernel_performance | 241600 | 9 | 2.4M | 0.0 | Reg. | Computing |
| 44089 | credit | 16714 | 10 | 184K | 0.0 | Class. | Financial/demographic |
| 44122 | pol | 10082 | 26 | 272K | 0.0 | Class. | Industrial/operational |
| 44136 | wine_quality | 6497 | 11 | 78K | 0.0 | Reg. | Human behaviour |
| 44137 | Ailerons | 13750 | 33 | 468K | 0.0 | Reg. | Other or not provided |
| 44145 | sulfur | 10081 | 6 | 71K | 0.0 | Reg. | Other science |
| 45020 | default-of-credit-card-clients | 13272 | 20 | 279K | 0.0 | Class. | Financial/demographic |
| 45022 | Diabetes130US | 71090 | 7 | 569K | 0.0 | Class. | Medical/human sensor |
| 45026 | heloc | 10000 | 22 | 230K | 0.0 | Class. | Financial/demographic |
| 45032 | yprop_4_1 | 8885 | 42 | 382K | 0.0 | Reg. | Medical/human sensor |
| 45038 | road-safety | 111762 | 32 | 3.7M | 0.0 | Class. | Human behaviour |
| 45039 | compas-two-years | 4966 | 11 | 60K | 0.0 | Class. | Human behaviour |
| 45041 | topo_2_1 | 8885 | 255 | 2.3M | 0.0 | Reg. | Medical/human sensor |
| 45043 | seattlecrime6 | 52031 | 4 | 260K | 0.0 | Reg. | Human behaviour |
| 45045 | delays_zurich_transport | 5465575 | 11 | 66M | 0.0 | Reg. | Industrial/operational |
| 45046 | Allstate_Claims_Severity | 188318 | 124 | 24M | 0.0 | Reg. | Industrial/operational |
| 45047 | Airlines_DepDelay_1M | 1000000 | 5 | 6.0M | 0.0 | Reg. | Industrial/operational |

## B.1 Contamination Analysis

To ensure that the datasets used for training do not contain any information about the evaluation data, we extract a range of metadata from each dataset and compare them across all pairs of training and evaluation datasets. This includes: (i) dataset names, (ii) hashes of dataset files, (iii) numbers of columns and rows, (iv) target mean and variance, (v) mean, variance, skew, and kurtosis of each feature, and (vi) coefficients of a univariate linear fit between each feature and the target if available. To allow for efficient pairwise comparisons between all features in all datasets, we use $k$-d trees [3] constructed for each dataset that contain the feature statistics. Any dataset pairs with unusual similarities were manually evaluated and those found to be related were removed from training.

## C Model Architecture and Hyperparameters

### C.1 Architecture Details

The model architecture (see Figure 2b) is comprised of input embedding functions, multiple transformer encoder layers, and task-specific output heads. The key architectural parameters are summarized in Tables C.1 and C.2.

**Preprocessing** We deliberately do minimal pre-processing of the data to ensure that our approach has wide applicability. All columns containing non-numerical values are mapped to integers using scikit-learn's [49] LabelEncoder function. The table is then standardized to 0 mean and unit variance, and outliers beyond 10 are clipped. After retrieval, we obtain a local context $X_{ctx}$ and their labels $y_{ctx}$. $X_{ctx}$ is standardized before the forward pass to avoid distribution shifts, and $y_{ctx}$ is also standardized for the same reason if it is a regression target.

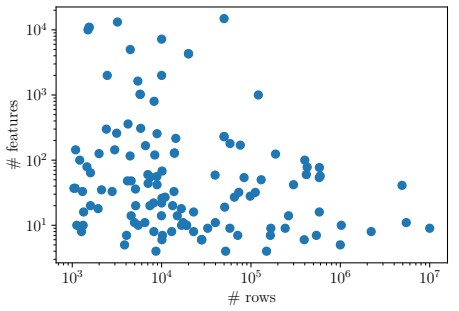
(a) Sizes of training datasets.

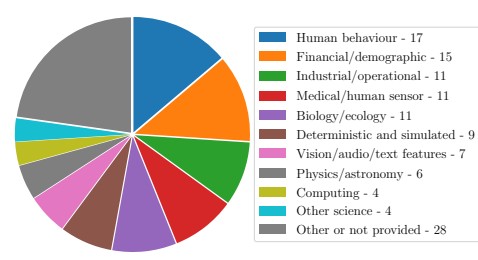
(b) Breakdown of training dataset domains.

Figure B.1: Breakdown of training datasets by size and domain.

Table C.1: Architectural Parameters

| Parameter | Value |
|---|---|
| Number of Attention Heads | 4 |
| Feedforward Network Factor | 2 |
| Maximum Number of Classes | 10 |
| Maximum Number of Features | 100 |
| Normalization First | Yes |
| Dropout Rate | 0.0 |

**Retrieval** We use the `faiss` library[3] for fast retrieval. All retrieval is done in the raw feature space after preprocessing, as in [58].

**Missing Value Encoding** We experimented with several strategies for missing value handling, including concatenating a binary missing-or-not mask, however the improvement was minimal at nearly double the compute cost. Hence, we opt for a simple strategy to zero out missing values and let the model learn how to deal with incomplete inputs. Note that zeroing out is done post normalization, meaning missing values are replaced with the mean.

**Optimizer** We use the Schedule Free optimizer from Defazio et al. [10] with AdamW [42]. We observed significant increase in performance and optimization speed compared to a cosine scheduler. Label smoothing and weight decay are applied throughout training and are important for smooth convergence. By default we set a learning rate of $5 \times 10^{-4}$ and weight decay of $5 \times 10^{-2}$ with label smoothing of $0.1$. The batch size is set to $256$ and both context and query lengths are set to $1024$. Model parameters are kept in brain float 16-bit (`bfloat16`) format.

# D  Pseudo-Code for Training Algorithms

In this section, we list the pseudo-code for our training procedure. In Code Block 1, we show the PyTorch `Dataloader` component. In the initialization phase, we first process the downloaded data and features by filling in missing values with the mean column values and creating a `faiss` index for fast retrieval. Next, in each worker within the `getitem()` function, we sample a random dataset, then we sample a random query within the dataset. After that, we mask out the target column and retrieve its approximate neighbours. Then we process the features and targets by random sub-sampling and random partitioning.

In Code Block 2, within each training step, we partition both the data X and targets y into context and query points by sampling an integer uniformly from 10 to its total length (inclusive of start point but exclusive of endpoint). We call this random evaluation position `eval_pos` in the code block. The points to the left of the evaluation position are then taken as context (i.e., `y_ctx`), and the points to the right of the evaluation position are taken as queries (i.e., `y_qy`). Finally, we calculate the appropriate loss depending on the task and optimize the network.

---

[3]`https://github.com/facebookresearch/faiss`

Table C.2: Number of Layers and Transformer Dimensions

| Number of Layers | Transformer Dimension |
| --- | --- |
| 3 | 32 |
| 4 | 64 |
| 5 | 96 |
| 6 | 256 |
| 10 | 384 |
| 12 | 512 |
| 16 | 768 |

Code Block 1: Pytorch Dataloader

```python
from torch.utils.data import Dataset
import numpy as np
import random

class TrainingDataset(Dataset):
    def __init__(self, dataset_ids):
        self.datasets = []
        for dataset_id in dataset_ids:
            X <- download dataset using dataset_id
            X <- process features of X (handle missing values,
                scale)
            knn_index <- compute knn index using FAISS
            self.dataset.append([X, knn_index])

    # Random column subsample and shuffling
    def create_random_columns(self, X):
        N, F = X.shape
        num_features_sampled = random.randint(F // 2, F)
        random_features_indices = np.random.choice(F,
            num_features_sampled, replace=False)
        return X[:, random_features_indices]

    # Generate a random classification or regression target for
        training
    def generate_random_target(self, y, cls_threshold=10):
        if len(np.unique(y)) > cls_threshold:
            # if there are more than 10 unique values in the
                target, we keep it as regression 70% of the time
            if np.random.rand() > 0.3:
                return y, "regression"
            else:
                # sample a random number of classes by binning
                    and divide into classes
                num_class = np.random.randint(2, cls_threshold)
                cls_boundary = np.random.choice(sorted(np.unique
                    (y))[1:-1], num_class-1, replace=False)
                y = (y[:, None] > cls_boundary[None, :]).sum(1)
                y <- label encode, shuffle y
                return y, "classification"
        else:
            assert len(np.unique(y)) > 1
            y <- label encode, shuffle y
            return y, "classification"

    # Generate a sample for retrieval
    def __getitem__(_):
        # sample a random dataset
        sample_id = np.random.choice(len(self.dataset), 1)[0]
        X_sample, knn_index_sample = self.dataset[sample_id]
```

```
44          N, F = X_sample.shape
45
46          # sample a random query from the dataset
47          x_q = X_sample[random.randint(0, N-1)].copy()
48
49          # sample a random column to be the target
50          target_idx = random.randint(0, F-1)
51
52          # retrieve approximate neighbours using x_q with
                target_idx masked
53          x_q[:, target_idx] = 0
54          X_nn <- find k neighbours using knn_index_sample with
                x_q as query
55          y_nn = X_nn[:, target_idx]
56          X_nn = np.delete(X_nn, target_idx, axis=1)
57
58          # subsample and shuffle features
59          X_nn = self.create_random_columns(X_nn)
60
61          # generate random target and task
62          y_nn, task = self.generate_random_target(y_nn)
63
64          return X_nn, y_nn, task
```

Code Block 2: Training Loop

```
1
2   model = Transformer()
3   optimizer = schedulerfree.AdamWScheduleFree()
4
5   for epoch in range(num_epochs):
6     model.train()
7     for X, y, task in train_loader:
8         eval_pos = random.randint(10, len(y))
9         y_ctx, y_qy = y[:eval_pos], y[eval_pos:]
10        y_ctx = zero_pad(y_ctx, N_qy, dim=1)
11
12        output = model(torch.cat(X, y_ctx))
13
14        if task == "classification":
15            loss = cross_entropy_loss(output, y_qy)
16        elif task == "regression":
17            loss = mse_loss(ouput, y_qy)
18
19        opitmizer.zero_grad()
20        loss.backward()
21        optimizer.step()
```

# E   Elo and Glicko2 Ratings

We expand the pairwise method comparison with Elo [14] and Glicko2 [20] ratings. For the Elo calculation, we estimate uncertainty by bootstrapping over match order permutations [6]. Glicko2, on the other hand, provides uncertainty by design and is less sensitive to match order in our experiments.

In Figures E.1a and E.1b, we report the Elo and Glicko2 scores respectively. The results are consistent between the two plots, with TabDPT performing best on both metrics, followed by the leading TFM baseline TabPFN v2.

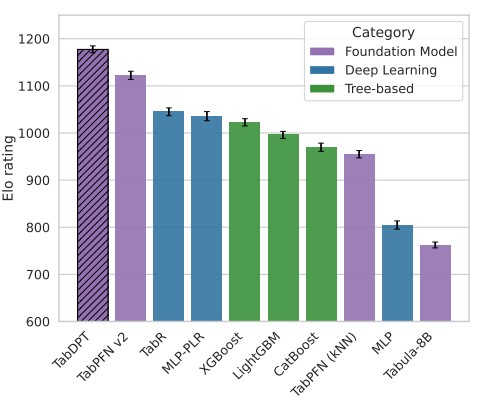
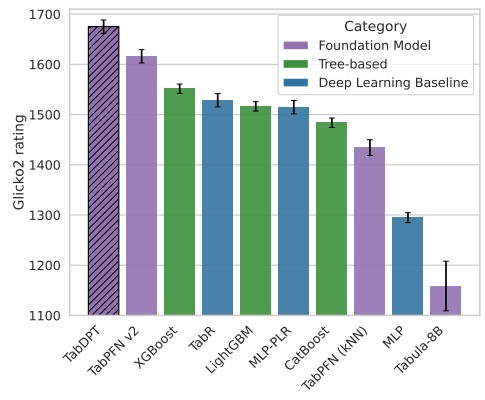

(a) Elo scores (Accuracy, $R^2$) with error bars.     (b) Glicko2 scores (Accuracy, $R^2$) with error bars.

Figure E.1: Duel-based metrics computed on accuracy and $R^2$ scores. (a) Elo ratings. (b) Glicko2 ratings.

# F Additional Results

## F.1 Additional Results by Dataset Statistics

In this section, we analyze the performance of all methods, bucketed by different characteristics of the benchmark datasets. In particular, we analyze performance by number of rows, number of columns, categorical fraction, and missing fraction. We see that TabDPT is robust across various dataset characteristics, with a very slight relative decrease in performance for very large CC18 datasets; this can be mitigated by fine-tuning as suggested in Thomas et al. [58].

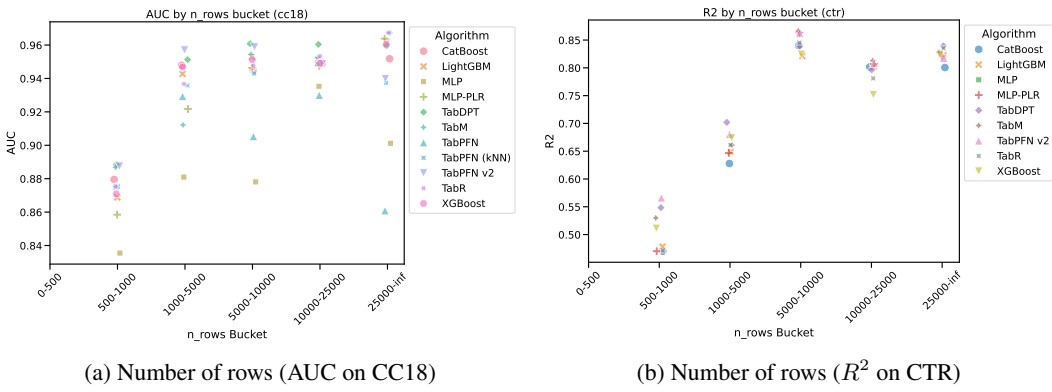

(a) Number of rows (AUC on CC18)  (b) Number of rows ($R^2$ on CTR)

Figure F.1: Comparison for Number of Rows.

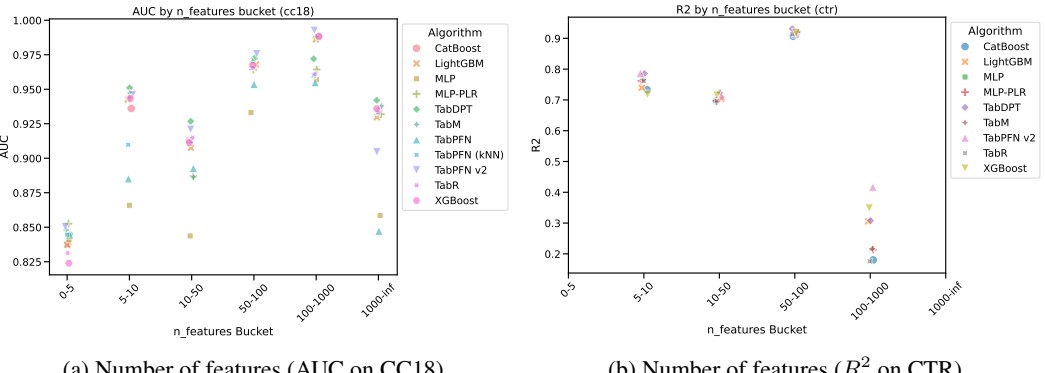

(a) Number of features (AUC on CC18)  (b) Number of features ($R^2$ on CTR)

Figure F.2: Comparison for Number of Features.

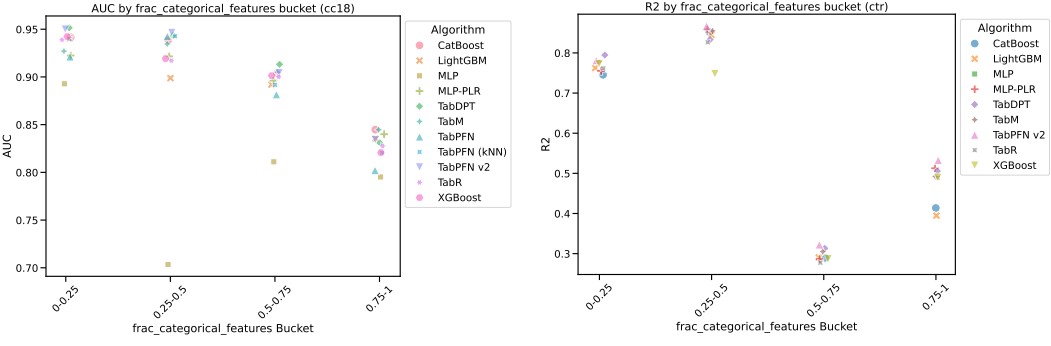

(a) Fraction of categorical features (AUC on CC18)  (b) Fraction of categorical features ($R^2$ on CTR)

Figure F.3: Comparison for Fraction of Categorical Features.

# G   Critical Difference Diagrams

Critical difference diagrams computed over CC18 and CTR23 fold results are given in Figures G.1, G.2, G.3, and G.4. The difference in confidence intervals compared to Table 1 are due both to how we compute the ranks and their confidence (using IQM, based on the recommendations from [1]) which computes uncertainty over different realizations of the suite. In contrast, the critical diagrams compare the uncertainty, with multiple hypothesis testing adjustments, over individual dataset realizations.

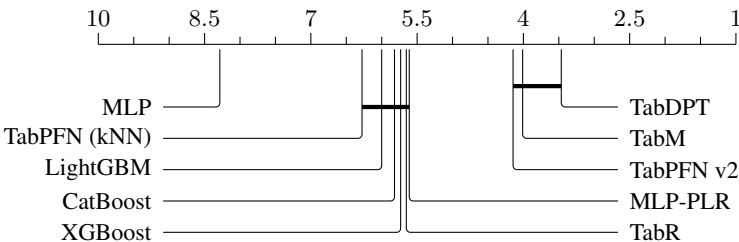

Figure G.1: AUC critical difference diagram

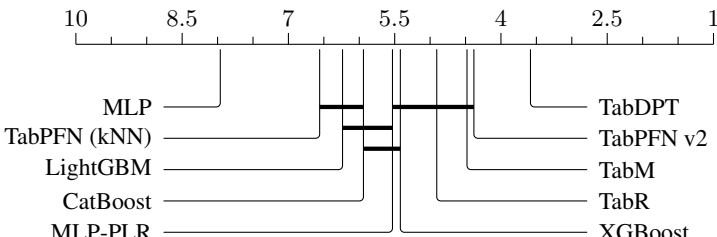

Figure G.2: Accuracy critical difference diagram

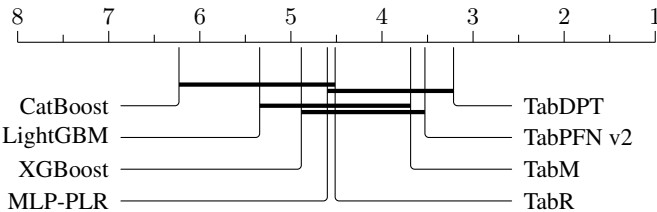

Figure G.3: Correlation critical difference diagram

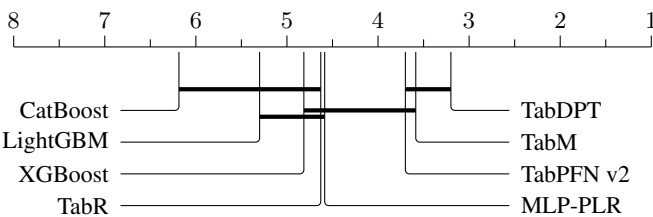

Figure G.4: R2 critical difference diagram

