# OpenReview forum: "TabDPT: Scaling Tabular Foundation Models on Real Data"
_NeurIPS.cc/2025/Conference — NeurIPS 2025 poster_

### Official Review · Reviewer_WEPY · 2025-06-26

**Clarity:** 4
**Significance:** 3
**Originality:** 3
**Rating:** 5
**Confidence:** 4

**Summary:**

The paper introduces TabDPT, a tabular foundation model that has been pre-trained on (real) datasets from the internet. TabDPT leverages self-supervised learning and certain dataset-level augmentations to extract more information from available datasets, and the paper shows that its pre-training follows scaling laws for data size and model size. TabDPT also includes context retrieval allowing it to scale to larger datasets. The paper includes experiments on different benchmarks, where TabDPT is the best-performing method for classification and regression, and for few-shot learning.

**Questions:**

- See the questions in the weaknesses part.
- Does TabDPT use ensembling of different shuffles/settings like TabPFN?
- What type of head and loss do you use for regression? Is it the same strategy as TabPFNv2?
- In the few-shot evaluation, why did you only evaluate 7 of the 8 dataset of the "STUNT" paper? Is the "income" dataset included in the pre-training?

**Minor comments**:
- l. 96: "another concurrent method requires a complex, three-stage procedure to learn from synthetic generators [47]" - you make it sound like the three-stage procedure is required because of the synthetic generators, when in fact the three-stage procedure is used to efficiently scale to larger context windows, regardless of the type of data used for pre-training.
- l. 159: what if you pick a categorical column but still want to do regression?
- l. 329: "suing" -> "using"

**Conclusion**:
The paper brings interesting contributions to the community, and if the concerns about evaluation fairness and accuracy of claims can be sufficiently resolved, I am willing to recommend acceptance.

**Ethical Concerns:**

["NO or VERY MINOR ethics concerns only"]

**Final Justification:**

- Evaluation concerns were mostly addressed: Baselines are not run with cross-validation, but the authors agreed to include foundation models with both train+validation data and train data only, to illustrate the difference. Other than that, the authors included TabM into the evaluation.
- Generally, I think the paper contains interesting insights on pre-training with real data and scaling laws, as well as on how to scale foundation models to large datasets, and should therefore be accepted.
- The model is probably not state-of-the-art in terms of benchmark scores when using the strongest baselines, but still strong at least for regression, and relatively fast when inference time is not a big concern.

**Limitations:**

Some limitations of the model are discussed at the end of the main paper, but limitations of the evaluation methodology are not discussed. I would argue that the latter should be discussed.

**Quality:**

3

**Strengths And Weaknesses:**

**Strenghts**:
- The paper is well-written.
- The paper provides interesting experiments on a few aspects of pre-training.
- The paper introduces some new and original ideas on pre-training with real-world data and other aspects.
- Comprehensive code is provided (though the evaluation of baselines seems to be missing).

**Weaknesses**:
- Details on the benchmarking setup are missing, like: How are training+validation sets generated and used by different methods? What search spaces do these methods use and for how many steps are they tuned (specifically, when do you use the settings from TabZilla and when the ones from TabR, regarding search spaces, tuning method, number of tuning steps, etc.)? How are TabPFNv2 and TabDPT evaluated (does it get train+validation data or only train data)? How are large datasets (> 10K samples) handled for TabPFNv2?
- From what I can tell, the comparison of TabDPT to baselines is not fair. Specifically, TabDPT can use the full training+validation set as context, while TabZilla uses holdout validation for baselines. However, the baselines could be much better by using cross-validation, which has even been explicitly demonstrated for TabZilla (https://arxiv.org/abs/2503.09159). I understand that many benchmarks use holdout validation because cross-validation is expensive, however, in this case it would be "more fair" in my opinion to only give the foundation models access to the training set and not the validation set, since it gives them a similar disadvantage to the one that other models have by not using cross-validation.
- The paper seems to overclaim its results considering the weaknesses in the evaluation. Generally, the new TabArena benchmark (https://arxiv.org/abs/2506.16791) evaluates TabDPT and comes to quite different results than this paper. I know that the benchmark was uploaded after the NeurIPS deadline, so I am definitely not suggesting that the authors should have known about it when writing the paper. Nonetheless, I think the benchmark demonstrates that if the authors had used best practices that were known for a while (cross-validation, weighted ensembling, more tuning (?), more recent baselines like RealMLP/TabM/TabICL), they would have come to the conclusion that their method is not state of the art. I know that the paper has been on arXiv for a while and I don't mean to demand that all of the newest baselines be included, but given these circumstances I think the paper should be more modest about "SOTA" claims. Perhaps including the TabArena results would be the easiest way for the authors to address the concerns about the evaluation.
- The retrieval of an individual context per sample does not allow for "TabPFN-style batching", making inference very slow (and this is not properly discussed / ablated in the paper - Section 4.7 is about training+inference, not just inference). (Adding to this comes the fact that model selection usually requires predictions for the validation set(s), which also take long to generate for methods with slow inference.)
- The paper shows that TabDPT outperforms TabPFNv2, but it is unclear if this actually holds because of the retrieval procedure for TabDPT that presumably allows it to perform better for large datasets, or if TabDPT is also competitive with TabPFNv2 in the regime that TabPFNv2 is actually designed for.

**Minor weaknesses**:
- l. 73: You claim "We show that applying our pre-training procedure to real data leads to faster convergence and better downstream accuracy than using purely synthetic data.", but you only show it for one specific way of generating synthetic data (the TabPFNv1 way). Please don't overgeneralize the results.
- The many-class classification approach in l. 207 is not ablated vs. some related approaches (e.g. TabPFNv2 extensions or TabICL) and it has the weakness that it can only model a certain factorized probability distribution over classes, instead of arbitrary probability distributions. Hence I would expect that this approach should not work well for probability-based metrics like log-loss or Brier loss. It might be good to ablate it or at least disscuss the potential advantages/disadvantages.

---

> ### Author Rebuttal · Authors · 2025-07-31
>
> We appreciate the time you have taken to thoroughly review our work, and are happy that you find it to be well-written, interesting, and novel. We respond to your comments below.
>
> ### Weakness 1: Details on Eval Protocol
>
> We used the datasets and splits from TabZilla: each dataset is divided into 80% training, 10% validation, and 10% testing set for up to 10 folds. For tree-based baselines (and kNN/MLP), we use the results computed by TabZilla: they run 30 trials on each fold.
> For TabR and MLP-PLR, we used their official repos and pre-defined search space, also using a 30-trial HPO.
>
> TFMs such as TabDPT and TabPFN tend to use both the training set and validation set for predictions. To address your point, we re-ran TabDPT only using the training set. This led to a small decrease in performance where, excluding TabPFN with validation in the context, TabDPT was still the best performing method, but not significantly above TabR.
>
> However, all of our non-TFM baselines extensively use the validation set to improve performance through their HPO, so it would be unfair to not use the validation set at all in our method. Incorporating all available data in the context is the most straightforward way for us to leverage the same amount of data that these training and tuning procedures are using.
>
> For TabPFN v2 we tried several things, including subsampling first to 10k rows or using ignore_pretraining_limits=True and using up to 30k rows. The latter experiment did not result in significantly different results and failed on a couple datasets. TabPFN v2 actually performs subsampling to 10k rows when using this option as can be seen in issue #169 on their repo in a comment from Noah. We also tried RandomForestPFN from tabpfn-extensions, but it resulted in segmentation fault errors on large datasets.
>
> Finally, we have now run TabM as a baseline and included the results in the rebuttal to Reviewer `ggCD`.
>
> ### Weakness 2: Concurrent Work on Best Practices
>
> Thank you for raising this - while we are aware of this paper and it has insightful observations, we would like to point out that it was published about 8 weeks before the abstract deadline and is thus concurrent. However, we are still happy to discuss and give more insight on our setup.
> First, the goal of our experimental methodology was to follow standard benchmarks to provide a fair and extensive comparison that was aligned with existing works, which CC18 and CTR23 allowed (instead of restricting the evaluation datasets to a subset on which our method would perform well). Our choice of CC18 and CTR23 (107 datasets) was also influenced by the fact that we could leverage the heavy HPO performed by TabZilla: 30 HPO rounds per fold and algorithm. We do not have the capacity in the short term to update our paper with a 5-split CV having 30-round HPO over 107 datasets and 2 folds for 5-6 baselines.
>
> **We would like to point out that many well regarded papers (such as TabM, TabR) also do not perform a full CV, but a holdout validation. Thus, we do still think that our baselines are legitimate, and will be more clear about the experimental setting in our paper.**
>
> ### Weakness 3: Concerns about Overselling
>
> Although we claim top performance on our benchmarks, the paper does not use “SOTA” or equivalent language to describe the model. If there are specific claims that you find are overstated though, we can moderate or clarify them.
> Regarding TabArena, we have been in contact with the authors and believe there are shortcomings in the evaluation of TabDPT that make it inappropriate for inclusion at this time.
>
> 1. TabDPT is one of the only models evaluated *without* tuning and ensembling, even though these could have been carried out in a similar way to TabPFN (and our performance is still competitive in this reduced setting).
> 2. The TabDPT version appearing on TabArena has `context_size=1024` and `n_ensembles=1` while the top performance we report here is significantly improved with `context_size=2048` and `n_ensembles=8`.
> 3. The metric used for multivariate classification expects a calibrated model, and our use of label smoothing during training substantially hurts this metric in particular compared to AUC/accuracy. However, it would be straightforward to calibrate our outputs (tuning temperature) for a fairer comparison.
> 4. **Comparison with other TFMs**. Note that the comparison with TabPFNv2 on the global leaderboard is not exactly fair: the datasets with >10 classes, >10k samples, or >500 features – which are exactly the ones where TabPFN can struggle with – were excluded from the pairwise comparison to compute Elo scores. This choice affects the model rankings in a manner favourable to TabPFN. Consider that TabDPT’s strongest aspect on TabArena is regression, where it ranks 3rd despite the points above. We could make up a rule that TabDPT is to only be used on regression tasks, which would practically ensure our model to be in the global top 3.
> 5. Different data distribution: This point is not a flaw but more concerning experimental design. We chose CC18 and CTR23 as our benchmarks, and the distribution of those datasets is different from the ones used in TabArena. For instance TabArena, on classification, tends to have larger datasets than CC18, which favors traditional methods that tend to scale better with the number of rows compared to TFMs. Furthermore, we give equal importance to regression and classification, but only about 25% of datasets within TabArena are regression tasks, which amplifies the bias of point 4 above.
>
> We are in contact with the TabArena authors to update the benchmark, and believe that our model will be competitive with top models once issues 1-3 are addressed.
>
> ### Weakness 4: Inference Speed Concerns
> We are happy to clarify some points in the paper. Retrieval is indeed a form of inference-time scaling where more compute is used at inference time in exchange for stronger performance.
> It is important to understand that our base architecture (rows as tokens) is very fast with TabPFN-style batching (~0.04s/1k samples), leading our architecture+retrieval batching to have a similar overall time to TabPFN v2 as their architecture (cells as tokens) is heavier.
> Despite this, our method performs very well given a time budget compared to traditional methods on CC18 and CTR23 because the training time is essentially 0.
> However we do agree that there are many scenarios (e.g., train once then deploy for years, or situations requiring hyper-fast inference) when inference-time becomes the bottleneck: in those cases, methods with very fast inference such as XGBoost could be preferred. We will add a discussion in the paper on this.
>
> ### Weakness 5: Scope of Comparison with TabPFN v2
> The fact that we achieve better performance on large datasets compared to TabPFN v2 is one of the main contributions of the paper, and one of the main diverging design choices setting us apart from them. In the Appendix (Figure E), we provide a breakdown by number of rows and features, and we have a performance similar to TabPFN v2 when rows <10k.
> Generally, we appreciate that different models in this space will have different strengths depending on data characteristics. We use CC18 and CTR23 to provide a standardized overall picture of model performance on a variety of data, and we think that showing effective performance on this set of datasets is useful, but we agree that we would expect to be better than TabPFN on certain kinds of data and worse on others.
>
> ### Minor Weaknesses
>
> **mW1**: Thank you for the catch, we can update the paper to be more specific about which generator is used. Note that most TFMs we are aware of were trained on variants of this particular synthetic data generator [3,4,5,6,7].
>
> **mW2**: Yes this is true. We chose this method as it was the one requiring the least amount of forward passes we could think of and it still performed decently well. [2] actually analyzed different multiclass methods including this one and showed that alternative methods can perform better.
> We generally kept all aspects of the model and pre-processing as simple as possible but we are excited to try other simple post-training methods which could improve our performance.
>
> ### Responses to Questions
>
> **Q2: Ensembling**
> We accidentally omitted this from the paper - we apologize for the oversight and will update it: yes, we evaluate using an ensemble of 8 predictors to match TabPFN, randomizing input feature ordering.
>
> **Q3: Regression Head**
> We used a simple scalar head with a L2 loss. The regression strategy for TabPFN v2 is not public, but, while we don’t know the exact details, it is certainly different: they use a (soft?) binning strategy with a head of high dimension (in the 1000s) which can then be mapped to a scalar. We can assume that they use something akin to the “histogram loss” [1] or other type of “regression as classification” strategy.
>
> **Q4: STUNT**
> Yes, `income` was incorrectly labelled as being part of CC18 in the STUNT paper; it is actually not and we used it in our pre-training set. Note that we did not observe overfitting on our pre-training set because of all the transformations done with SSL, but we chose to strictly exclude any dataset used in pre-training from evaluation.
>
> [1] “Investigating the histogram loss in regression”, Imani et al., 2024
>
> [2] “A Closer Look at TabPFN v2: Understanding Its Strengths and Extending Its Capabilities”, Ye et al., 2025
>
> [3] “TabPFN: A Transformer That Solves Small Tabular Classification Problems in a Second”, Hollman et al., 2023
>
> [4] “Accurate Predictions on Small Data with a Tabular Foundation Model”, Hollman et al., Nature 2025
>
> [5] “TabICL: A Tabular Foundation Model for In‑Context Learning on Large Data”, Qu et al., 2025
>
> [6] “Attic: A New Architecture for Tabular In‑Context Learning Transformers”, den Breejen et al., 2024
>
> [7] “TabFlex: Scaling Tabular Learning to Millions with Linear Attention”, Zeng et al.,2025

---

> > ### Comment · Reviewer_WEPY · 2025-08-01
> > **Rebuttal resolves many points except cross-validated baselines**
> >
> > Thank you for your detailed response. I assume you will include some of the clarifications of the response in the paper (such as the details on the evaluation).
> >
> > Weakness 1/2:
> > - I agree that a comparison using only the train set for TabDPT is not fuly fair either, but I am not sure in which direction. Let's call this setting A and the ideal comparison of cross-validated baselines to TFMs with full data setting B. In my experience, for default parameters, setting A and B give roughly similar results in terms of the relative comparison between methods, even though baselines can use the validation set for early stopping (with most NNs being worse in setting A due to the lack of ensembling). I don't know how this changes with hyperparameter tuning for baselines - one might expect this to favor them in setting A compared to B, since they can use the validation data even more while TFMs cannot, but actually it might make them worse in setting A compared to B because hyperparameter optimization overfits much more easily without cross-validation (and especially with a val size of only 10%, see again https://arxiv.org/abs/2503.09159). Since you mentioned that TabDPT would only be slightly better than TabR in setting A, I would assume it is worse than TabM then?
> > - Regardless of whether A is fair or not, my point still holds that the current evaluation is unfair - consider, e.g., that the comparison in TabArena would be much different if all the non-TFM baselines used holdout validation (Figure 5 right in the TabArena paper, which uses tuned+ensemble but should still be sufficient to make the point). I understand that it is infeasible to run cross-validation experiments in a short amount of time. However, I still think that it is a weakness that should be addressed in some form. How do you plan to adress it?
> > - I don't think the argument that many well regarded papers such as TabM, TabR use holdout validation is a good one, since these papers use holdout validation for *all* methods which is more fair (they don't have TFMs in their comparison).
> >
> > - (Minor) re #169: from the comment it is not clear to me if the subsampling only happens for fitting the preprocessing. Indeed, that would be more plausible, given that 30k failed for you on some experiments.
> >
> > Weakness 3: Thank you for the clarification, these are (mostly) good points:
> > 3. Good point, although I expect temperature scaling to be suboptimal since it is linear in logit space, and the optimal logits for label smoothing are a nonlinear transform of the true logits (probabilities near 0 or 1 get squashed together in logit space). This squashing might also hurt AUC (as in the RealMLP paper) - the results on binary classification with AUC in TabArena are still quite bad, but maybe the ensembling and larger context in your paper fixes it.
> > 4. I assume you refer to the TabPFNv2-specific leaderboard. Anyhow, my point is not about being SOTA on specific leaderboards, but about the disagreement beetween results, which you have addressed.
> >
> > Regarding overselling, while you do not literally claim "SOTA", some claims are pretty close to it, e.g.:
> > - "Our resulting model, TabDPT, achieves top performance on both regression (CTR23) and classification (CC18) benchmarks."
> > - "TabDPT shows the best overall performance across all models."
> > - "TabDPT also significantly outperforms both deep learning and tree-based methods that are trained for each dataset".
> >
> > My concern is that these claims may not hold up with a proper cross-validation of the baselines.
> >
> > Weakness 4: Thank you for the discussion. You might want to mention whether you ran models with AMP or not (if you haven't already).
> >
> > Weakness 5: Thank you for pointing to Appendix E, this resolves my point.
> >
> > Minor weaknesses: Thanks, this addresses my points.
> >
> > Q2: TabPFNv2 uses 4 and not 8 ensemble members by default, did you change the value to 8?
> > Q3: AFAIK the regression strategy for TabPFNv2 is known and is equivalent to what they used in the PFNs4BO paper, which is open source (and maybe also the original PFN paper (not TabPFN)).
> >
> > Since the concerns have partially been addressed, I am raising my score from 3 to 4.

---

> > > ### Author Response · Authors · 2025-08-06
> > > **Discussion on paper changes and TabPFN v2 details**
> > >
> > > We sincerely thank you for your engagement and helping us improve the paper.
> > >
> > > Thank you for sharing your qualitative experience; you have made good points. We agree that devising fair comparisons between TFMs and non-foundational techniques is an important discussion which will permeate future research on TFMs. Since TFMs don’t suffer as much from overfitting, they might benefit from a different usage of the validation set (concatenation with training set), and so it is hard to know how to best compare them to non-foundational methods.
> > >
> > > These two classes of methods are also used for different purposes, and so while we could technically also fine-tune using the validation set or perform cross-validation with TabDPT (see for example [1] which obtained a significant performance improvement using early stopping on validation set), as far as we know, this does not align with the popular goal of getting decent and fast predictions on a new task for TFMs.
> > >
> > > Yes, TabM (with 30 HPO trials) is better than TabDPT (no validation): this is significant on classification metrics (non-overlapping CI), but not significant on regression metrics (overlapping). It is however worth reiterating that training and tuning TabM for individual problems is much slower, as it took roughly 30 times longer to achieve these results when compared with the inference time once we have a pre-trained TabDPT available with the heaviest setting. In any case, we will note this result in the final version of our paper.
> > >
> > > We are open to suggestions you may have to do this, and overall improve the experimental section, providing one actionable proposal: we can update the main table with the no-validation results for TabDPT and TabPFN v2.  We will cluster those with all the baselines results and separate the train+valid results.
> > >
> > > We would bold the best results for “validation set fair” setting, so TabM would be bolded alone (depending on TabPFN no-val) on classification but would be bolded alongside TabDPT on regression. TabDPT (train+val) would be underlined.
> > > We would provide a discussion and citation to the “Unreflected [..]” paper we discussed and explain that hold-out validation is probably more directly comparable to TabDPT/TabPFN (no val). However we do also report the train+val numbers as they can be obtained with basically no additional effort compared to the no-val setting, while CV might be more expensive.
> > > We can furthermore add a table/figure with pure inference times; here tree-based methods would shine and TabPFN/TabDPT would be the slowest. Another reviewer also asked for this and we are happy to oblige.
> > > We can also be careful with language, ‘among the top’, ‘best on metric X and dataset Y for setting Z’, etc.
> > >
> > > Would this alleviate some of the concerns you had?
> > >
> > >
> > > **W3:** Thank you for the precision and the very insightful comments. You are right that we can’t expect tuning a scalar on the logits to fix everything on all metrics. However, given how under-confident the model predictions are, we have strong evidence that using a temperature < 1 would result in better cross entropy across the board. Furthermore, now that we know that cross entropy is one of the metrics considered, we can also remove or reduce the label smoothing: in particular, we have now achieved improved training stability with our modified transformer block (adding QK normalization and an additional layer norm), so it should be possible for us to reduce label smoothing now.
> > >
> > > **W4:** Thank you, yes we use AMP, we can expand on this as well.
> > >
> > > **Q2:** From what we can see, the default hyperparameter for n_estimators is 8 in the TabPFNClassifier class (line 143 of `classifier.py` in the TabPFN repo); we can rerun it with 4 as well. Note that the time vs. performance figure would still allow us to compare models given a similar budget even if the n_estimators differ.
> > > Concerning regression something of the sort would be our guess as well, but we talked to some of the authors and the details appeared to be private.
> > >
> > >
> > > [1] Thomas V, Ma J, Hosseinzadeh R, Golestan K, Yu G, Volkovs M, Caterini AL. Retrieval & fine-tuning for in-context tabular models. Advances in Neural Information Processing Systems. 2024 Dec 16;37:108439-67.

---

> > > > ### Comment · Reviewer_WEPY · 2025-08-07
> > > > **Cross-validation issue (mostly) resolved**
> > > >
> > > > I like your proposals for addressing the cross-validation issue - I think this is basically the best you can do without running the cross-validated baselines. I will raise my score to 5.
> > > > (The time to reach the first inference is a good motivation - one could argue that the time gap would reduce with meta-learned portfolios as in AutoGluon and (esp. for tree-based models) with parallelization. The portfolio thing could also be done with TabDPT once a tuning space is available, of course.)
> > > >
> > > > W3: sure, I expect a temperature < 1 to improve things (even TabPFN already uses 0.9). And good to hear that you are seeing improved stability without label smoothing.
> > > >
> > > > Q2: The default n_estimators of TabPFN was recently changed from 4 to 8, in this commit, which seems to belong to v2.1.1, and should therefore have been 4 in the version you used in the submission: https://github.com/PriorLabs/TabPFN/commit/eb08e94218f8b2aed3a5eae0588f9140361ab5bb
> > > > (My intention here is primarily to make sure that you correctly describe the parameter that was used in TabPFNv2, though I also think an evaluation at the same number of estimators would be preferable.)

---

> ### Author Response · Authors · 2025-08-08
> **Answer to issues mostly resolved**
>
> Thank you for the advice and reassessing your score! We'll make sure sure to include these more nuanced takes in the final version.
>
> W3: To be exact we can reduce label smoothing by a significant amount, but we had not considered it a priority until now. Note that in our SSL data we can generate combinations where some classes only appear in the queries and not in the context, we think this is a reason why we still need some amount of label smoothing. We will investigate if fixing this can allow us to remove label smoothing entirely (or to an even larger extent).
>
> Q2: We had not realized that TabPFN was updated to 2.1, with the continued pre-training using real data from what we gathered, we will make sure to include the version used in the paper, both for TabPFN and TabDPT (as we also plan to continue improving the model).
>
> Thank you again for engaging in the process and having a significant impact on improving our experimental section

---

### Official Review · Reviewer_4PRE · 2025-06-30

**Clarity:** 3
**Significance:** 3
**Originality:** 2
**Rating:** 5
**Confidence:** 3

**Summary:**

The paper proposes an in-context-learning based tabular foundation model. The model is trained using a self-supervised learning pipeline on real data, which is in contrast to close relatives such as TabPFN that are trained on synthetic data. The authors claim that models trained on real data lead to significantly faster training and better performance. They also present tabular foundation model "scaling laws", which characterize how scaling model size and training set size leads to better performance.

**Questions:**

Please see the Strengths and Weaknesses section for questions/suggestions.

**Ethical Concerns:**

["NO or VERY MINOR ethics concerns only"]

**Final Justification:**

I acknowledge the author's rebuttal, which addresses most of my concerns. I'm not convinced with the authors' argument that a model trained on real data outcompetes a model trained solely on synthetic data -- not that this cannot be true, but because there is not enough evidence in their paper to show that this is the case. Though not without its flaws, this paper is a good contribution and I would like to see it published.

I have raised my score following the reviewer's response to my comments, with the assumption that they will modify the text for the final version as described in their response.

**Limitations:**

Yes

**Quality:**

3

**Strengths And Weaknesses:**

Strengths: The paper takes a step towards building fast and generalizable algorithms for predicting small tabular datasets. The paper is valuable as an exploration of the TFM space, and in particular, focuses on the value of training a TFM using real datasets.

Weaknesses: In my opinion, there are two major weaknesses of this paper.

First, there are various issues with how this paper is framed. For instance, the abstract can be interpreted as being somewhat misleading. The authors state that "In this work, we propose an approach to combine ICL-based retrieval with self supervised learning to train tabular foundation models.". There have been previous such approaches, some of which are cited by the authors in the main text, most notably TabPFN and its latest variant TabPFN v2. It would help the authors' case if they did *not* motivate their work using statements such as "This direction is gaining popularity but is still in early stages with relatively few tabular-based TFMs developed" (lines 50-51), and rather precisely state what their paper contributes relative to what is already out there.

Other claims such as "In this paper, we hypothesize that real tabular data contains much more information than heavily engineered synthetic tabular generators, thus allowing more straightforward improvements by scaling model and data size, which is supported by experiments in Section 4.5." lack empirical support. Moreover, I found critical statements such as "However, they are completely reliant on synthetic data generators; ensuring that this mechanism captures the full diversity and nuances of real-world data is challenging, and making meaningful improvements to it is difficult." rather odd -- methods that perform close reliably while being trained only on synthetic data would in my opinion be considered advantageous as real data is hard to come by.

Second, I found quantitative comparisons between the different algorithms somewhat lacking. Averaging the rank (or the pairwise win-rate) over many datasets is a rather odd metric for comparison. The AUC is visible in Figure E, and it is clear that all models are quite close to each other. It is unclear if the authors have based their calculation of the average rank (in Table 1) based on a proper statistical comparison amongst the models. Can the authors please clarify? It would also be useful to plot the predictions from the model on some toy datasets to show that it generates reasonable predictions.

A main claim of the paper is that using real data is more efficient. The data that supports this claim is presented in Figure 4b -- however, it is unclear how much and what synthetic data is used. This is important as the test loss for synthetic data has not yet saturated.  The real value of using synthetic data is that a potentially limitless amount of it can be generated. A useful comparison between real data and synthetic data would show that the synthetic data generators that are being used fundamentally do not capture statistical features of real datasets even with infinite data. Does your model trained on a small, real dataset saturate at a smaller test loss compared to when it is trained on a large amount of synthetic data?

---

> ### Author Rebuttal · Authors · 2025-07-31
>
> We thank you for your review. Before addressing the weaknesses and questions, we would like to clarify a point you mentioned in the strengths:
> > The paper takes a step towards building fast and generalizable algorithms for predicting small tabular datasets
> One of our main goals with TabDPT is to generalize and scale to large datasets. Retrieval enables this by allowing us to keep fixed context size regardless of the dataset size. This is also the primary reason why our method tends to scale better as datasets become larger compared to TabPFN, as can be seen in Figure E.1.
>
> ### Weakness 1: Contribution
> > In this work, we propose an approach to combine ICL-based retrieval with self supervised learning to train tabular foundation models
>
> We will clarify this point, but the main claim here is related to our SSL approach that we use to train on real data. Most of the published TFMs such TabPFN, TabForestPFN, TabICL, TabFlex, and Attic [1-6] train on synthetic data, many of them on derivatives of the TabPFN v1 prior. We are the first, as far as we know, to have successfully trained an in-context TFM on real data by leveraging SSL.
>
> ### Weakness 1: Synthetic vs. Real Data
> This is an interesting discussion at the core of our approach and we think of it as a strength rather than weakness.
>
> It is true that given a synthetic data generator, infinite samples can be created from it. However not all synthetic generators are made equal. For instance, we had tried training a model with sklearn simple synthetic data (make_classification, etc.) by randomizing all the parameters of those generators. The resulting model performed well on those generators but did not generalize at all to real datasets. An improvement in the synthetic data generator is also an important part of the improvements from TabPFN v1 to TabPFN v2.
>
> Designing good synthetic data generators that lead to robust downstream generalization is challenging. The generator configuration files for v1.0.0/tabpfn/model_configs.py of TabPFNv1 and the priors in v1.0.0/tabpfn/priors show that extensive tuning was done to calibrate them.
>
> In comparison, our self supervised approach is simple, can be applied to any dataset, and importantly exhibits scaling laws with the number of datasets, indicating that we can continue to scale. Note that we only used 123 datasets, while the full Tablib [7] repository contains over 627M tables, meaning that there is a very significant amount of real data to still explore.
>
> All in all, we showed that a simpler approach using real data was able to match the best approaches trained on synthetic data. We showed that our approach scales as we increase the amount of data, and there is much real data available for pretraining. We do believe that this can lead to the same paradigm as in LLMs where obtaining a very large training corpus is of paramount importance. This can also be important for developing TFMs that can understand textual data.
>
> ### Weakness 2: Quantitative Comparison
> We aimed to provide different ways to understand the performance of the different models through pairwise comparison (win-rate, elo), ranks, and four different scores.
> For computing the scores and their confidence intervals we use the InterQuartile Mean and compute confidence intervals through bootstrapping with 20,000 repetitions. This was advocated in the NeurIPS 2021 work awarded as the best paper [8], which was concerned with how to best evaluate the performance of different algorithms on a suite of tasks with several seeds/folds per task.
>
> A few important things to note:
> A large reason for the high scores are the metrics used: ROC-AUC, accuracy, and correlation tend to have higher value. Reporting MSE for regression would lead to the same conclusion about the ordering of methods. For classification, one can use PR-AUC, which tends to be smaller, but we did not have access to it for the TabZilla baselines.
> Using the interquartile mean (which removes the top and bottom 25% values) also tends to lead to higher scores: with regular mean the scores are slightly lower (low 0.9s for best methods in AUC, high 0.8s for accuracy, low 0.8s for correlation and low/mid 0.7s R2 for best regression). Using the median for instance would result in even higher scores. This is because quite a few datasets in CC18 are easy to predict.
> Moreover, even though the scores are high, the differences between methods are statistically significant (see response to Reviewer `ggcD`).
>
> ### Weakness 3: Scaling and Synthetic Data
> At the outset, we clarify that the models have indeed converged in Figure 4(b), which we have confirmed by training up to 1024 epochs.
>
> That being said, this is still a very interesting question. While we have not included it in this version of the paper, we also have the performance of our model trained purely on synthetic data for different model sizes as well.
>
> Here are the main findings:
> For small models (30k parameters) we find that for the same number of steps trained, the synthetic data generator (TabPFN v1) outperforms models trained on real data.
> However, this changes as we increase model size, and eventually when using larger models real data becomes significantly better. We hypothesize that smaller models are not able to capture some complex and particular patterns that are present only in real data, but that larger models can capture these patterns.
> **However, as this observation may be specific to the particular synthetic data generator we used, we wanted to focus on the fact that we can reliably use real data, it works well, and is easy to scale, rather than a comparison that might become less insightful if new synthetic data generators are released.**
>
> | Training cells   |   33.87k |   158.54k |   423.18k |   3.46M |   12.48M |   26.35M |   78.09M |
> |:------------------|---------:|----------:|----------:|--------:|---------:|---------:|---------:|
> | Synthetic         |    0.636 |     0.594 |     0.582 |   0.569 |  div     |    0.563 |    0.565 |
> | 52.47M            |    0.656 |     0.630 |      div  |   0.589 |    0.586 |  div     |  div     |
> | 141.39M           |    0.668 |     0.633 |     0.604 |   0.581 |    0.575 |    0.571 |    0.574 |
> | 269.89M           |    0.648 |     0.612 |     0.591 |   0.572 |    0.564 |    0.563 |    0.566 |
> | 598.52M           |    0.641 |     0.594 |     0.577 |   0.567 |    0.563 |    0.562 |    0.563 |
> | 2.06B             |    0.643 |     0.597 |     0.580 |   0.564 |    0.559 |    0.556 |    0.554 |
>
> We present the table above as a recreation of Figure 1, but with synthetic data added.
> For all model sizes (trained for 512*128 steps with 256 batch size and 1024 tokens) the synthetic data thus consists of 16.7M tables, 17.7M rows, or about 858B cells. In comparison, our largest real data setting has 123 unique tables, 33M rows, and about 2B cells.
>
> [1] “TabPFN: A Transformer That Solves Small Tabular Classification Problems in a Second”, Hollman et al., ICLR 2023
>
> [2] “Accurate Predictions on Small Data with a Tabular Foundation Model”, Hollman et al., Nature 2025
>
> [3] “TabICL: A Tabular Foundation Model for In‑Context Learning on Large Data”, Qu et al., ICML 2025
>
> [4] “Attic: A New Architecture for Tabular In‑Context Learning Transformers”, den Breejen et al., 2024
>
> [5] “TabFlex: Scaling Tabular Learning to Millions with Linear Attention”, Zeng et al., ICML 2025
>
> [6] “Position: The Future of Bayesian Prediction Is Prior‑Fitted”, Müller et al., 2025
>
> [7] “TabLib: A Dataset of 627M Tables with Context”, Eggert et al., 2023
>
> [8] “Deep Reinforcement Learning at the Edge of the Statistical Precipice”, Agarwal et al., NeurIPS 2021 (Best Paper Award)

---

> ### Comment · Reviewer_4PRE · 2025-08-05
>
> Thank you -- I acknowledge the author's rebuttal, which addresses most of my concerns. I'm not convinced with the authors' argument that a model trained on real data outcompetes a model trained solely on synthetic data -- not that this cannot be true, but because there is not enough evidence in their paper to show that this is the case. Though not without its flaws, this paper is a good contribution and I would like to see it published. However, I will retain my score.

---

> ### Author Response · Authors · 2025-08-08
> **Answer to comment**
>
> Thank you very much for engaging with us! We are happy to hear that most of your concerns have been addressed.
>
> To give a bit more context about synthetic data vs. real data: at the core of the paper is the fact that we showed that we can indeed train on real data, that it scales, and provide a simple mechanism to do so.
> This is in contrast to most other approaches, which use synthetic data, so we need to address this and compare against it.
>
> Now, you are absolutely correct that we can't make a blanket statement that real data is objectively better than synthetic data. First, synthetic data can be hard to define in the first place; some synthetic data comes from models trained on real data, and even real data that is modified or augmented enough could be considered synthetic.
> Second, this is a burden-of-proof scenario where we cannot try all possible synthetic priors, for example, the TabPFN v2 prior, which is private.
>
> However, most recent Tabular Foundation Models (TabFlex, Attic, TabICL, etc...) use variants of the TabPFN v1 prior. In practice, many of the synthetic data generators that perform well share the same origins, so it is not unreasonable to group them under the broader label of “synthetic data,” as is common in the field.
> Furthermore, when comparing synthetic data vs. real data, synthetic has to be “engineered” in some way, i.e researchers have to come up with what they think are good priors. Maybe this will scale well in the future, but it is still unclear. In contrast, real data of varying modalities and complexities already exists.
>
> Finally, we do believe that synthetic data will have an important role to play and will continue to be used in Tabular Foundation Models. However, in this paper, we wanted to show that it is not only possible but also quite easy to train purely on real data and match or exceed the performance of similar models trained on synthetic data (i.e the broad family of synthetic data mentioned above).
>
> We can be more careful about our use of "synthetic data" and clarify what we mean by it exactly, would this alleviate your concern?
>
> Lastly, thank you very much for engaging in the process and raising good points, we will review some of the discussion we had to include in the paper to improve its clarity.

---

### Official Review · Reviewer_ggcD · 2025-07-02

**Clarity:** 3
**Significance:** 3
**Originality:** 3
**Rating:** 5
**Confidence:** 5

**Summary:**

The authors propose TabDPT, an ICL tabular foundation model that is trained on real datasets compared to other ICL counterparts by using SSL as a form of data augmentation. The proposed method is compared against relevant baselines on classification and regression benchmarks, where the proposed method achieves the best results. The authors ablate the different decisions/components of the proposed method.

**Questions:**

- Could the authors incorporate a retrieval based context for TabPFNv2 and analyze if the gains of TabDPT are over all datasets or only in larger ones.
- Could the authors provide critical difference diagrams to observe whether the difference in results is statistically significant.
- Moreover, the authors should correct Figure 3 b) in my perspective and provide the inference times for all methods, this would provide a greater insight for the domain practitioners.
- Do the authors include the retrieval context preparation in the inference time for TabDPT?
- It could be interesting to include TabM [1] in the results as a non-meta learned DL method, given that it claims state-of-the-art results and outperforms TabR.


[1] Gorishniy, Y., Kotelnikov, A., & Babenko, A. TabM: Advancing tabular deep learning with parameter-efficient ensembling. In The Thirteenth International Conference on Learning Representations.

**Ethical Concerns:**

["NO or VERY MINOR ethics concerns only"]

**Final Justification:**

The proposed work was solid to begin with.  The authors performed the following amends:

- The authors had a plot referenced as inference times which was in fact showing total time. The authors promised to amend it.
- The authors justified why one of the ICL baselines did not make use of a retrieval context.
- The authors provided information regarding the difference and whether it was statistically significant.
- Lastly, the authors added a recent non-meta learned baseline.

**Limitations:**

- The authors have adequately addressed the limitations. Although I believe a sentence about the higher inference time would be beneficial.

**Paper Formatting Concerns:**

I did not notice any issues with the formatting.

**Quality:**

3

**Strengths And Weaknesses:**

**Strengths**:

- The paper is well written.
- The proposed method is compared against relevant baselines on standard well-established benchmarks.
- The implementation and weights are provided.
- The authors provide ablations for the different design decisions.

**Weaknesses**:

- Figure 3 (b) reads as inference runtimes, however the authors include the train + hpo training time which is not fair for the non-meta learned baselines and confusing. I can also advocate that a non-meta learned model is only learned once per task and mostly then performs inference. I think it would be interesting to provide the inference times for all methods.
- The authors use a retrieval based context for TabPFNv1 but not for TabPFNv2. I think it is fair to use the same procedure for TabPFNv2. It can be that the proposed method only surpasses TabPFNv2 only for large datasets.
- The authors train the method on a large amount of real datasets and incorporate a considerable number of benchmarks, which makes it difficult to extensively evaluate the claims of the paper.

**Typos:**

- Line 329s suing -> using

---

> ### Author Rebuttal · Authors · 2025-07-31
>
> Thank you for taking the time and effort to provide such a thorough review, and we appreciate that your overall opinion of our work is positive. We now respond to your comments below.
>
> **W1 and Q3**: Yes, this is a valid concern, and we are open to being more clear about it in the paper. We do think that “time to prediction” on a brand new dataset, i.e., train+test time, is a very valuable metric, but so is pure inference speed in the cases where a model is trained once and meant to be used for a long time for inference (assuming there is no drift requiring frequent finetuning/retraining).
>
> Considering the tradeoffs in developing and evaluating a model can help motivate this metric. TabDPT inference time with retrieval is typically in the range 0.5s-3s/1,000 test points (depending on the context size and ensembling used). By comparison, XGBoost has an inference speed closer to 0.02s/1,000 test points so it is one to two orders of magnitude faster. However the cost of training, especially with HPO, is much higher. This leads to a tradeoff where, depending on the setting, specifically on the ratio between the number of test examples and train examples, foundation models like TabDPT or classical methods can be faster. On the `adult` dataset, as an example, a single XGBoost trains in 0.54s/1,000 samples. Note that XGBoost is also quite fast among tree based methods on CC18 and CTR23 as we found LightGBM (because it struggles on datasets with a large amount of features) and CatBoost to often be significantly slower for training. Furthermore, the TabZilla baselines included in our results perform a 30 HPO search per fold, which would result in about 30x the training time for the reported performance (which we believe is fair as we report the performance with HPO).
> To give an example, the heaviest setting for TabDPT (large context, ensembling) performs inference for CC18+CTR23 in about 1-2 hours. In contrast, we also tried to do the full HPO for LightGBM and this wasn’t finished after several full days of compute (while skipping/feature subsampling some large datasets).
>
> Other cases where the full time to prediction is a greater concern than just inference time include enabling models to be refit as new data comes in, efficient AutoML, one-off data analysis tasks, and rapid prototyping. While we don’t believe that our model or other ICL foundation models are ideal for all time, performance, and data size tradeoffs, we believe that our results still point to useful improvements on real world tasks.
>
> We intend to include an extended discussion in the paper as you raised an important and legitimate point. Let us know if you have any specific requests.
>
> **W2 and Q1**: This is a valid question, but this is where the design difference between TabDPT (rows as tokens) and TabPFN v2 (cells as tokens) becomes important.
> We chose to keep the model lightweight but much faster with the “rows-as-tokens” method, because retrieval adds a significant computation overhead (memory and time, because we now have a context for each test point) for higher performance. On small datasets, we do not perform retrieval but simply batch the examples with a shared context like TabPFN, which leads to about 0.04s/1,000 samples while on larger datasets the same setting (2048 context size, 8 ensembles) would be almost 100x slower.
> Now, given that TabPFN’s speed is comparable to the latter setting, adding a 100x factor could lead to an inference runtime for TabPFN v2 with retrieval on CC18 and CTR23 of about a week compared to about ~1h15/fold for TabDPT with heaviest settings. (This would occur because we would not share a context across queries, and it assumes an efficient batching implementation, which is itself not trivial because TabPFN already does optimizations such as KV caching the **single** context which would not necessarily translate well as it would have one context per sample.)
> Lastly, in Figure E.1 in appendix we provide a breakdown of the performance in function of the number of rows for CC18 and CTR23. While on classification TabDPT is only significantly better than TabPFN v2 for larger datasets (and tied otherwise) we believe this is already a valuable result as their inference times are comparable and one of our main concerns is the ability to perform well on large datasets.
>
> **W3**: We are happy to discuss any specific concern you may have, but we feel that training and evaluating using a wide range of real-world data is aligned with the goals of the paper. For instance, we think that our evaluation on many datasets is a strength, allowing us to compare performance across different dimensions (dataset size, number of features) as done in Figure E in the appendix. For clarity, we provide the list of all datasets used for training and testing in Appendix A. In case the concern is data leakage, we carried out detailed data leakage verifications described in Appendix A.1.
>
> **Q2**: This is a good idea. While we cannot share an image in this response we have performed a critical difference analysis for each metric. TabDPT has the highest rank for all metrics: accuracy, AUC, correlation, and R2. For accuracy, TabDPT is significantly different from all other methods, and for the other metrics, it is in a clique with one or more other method: with TabPFN for AUC; with both TabPFN and MLP-PLR for correlation; and with TabPFN, MLP-PLR, and TabR for R2.
> The difference in confidence intervals compared to Table 1 are due both to how we compute the ranks and their confidence (using `rliable`, based on the recommendations from the NeurIPS best paper award [1]) which computes uncertainty over different realizations of the suite.
> In contrast, the critical diagrams compare the uncertainty, with multiple hypothesis testing adjustments, over individual dataset realizations. Both methods are valid in our opinion and we are happy to include the critical difference diagrams in the paper.
>
> **Q4**: Yes. It is very small when compared to inference time. We also include retrieval time which is negligible. However if we need to use more complex (trained/fitted) indexing methods (HNSW…), which will be necessary on extremely large datasets, the index creation time and the retrieval can become more significant.
>
> **Q5**:
> > It could be interesting to include TabM [1] in the results as a non-meta learned DL method, given that it claims state-of-the-art results and outperforms TabR.
>
> Thank you for the suggestion. We have run a baseline using the TabM official implementation both with default settings and 30 rounds of HPO (with the search space used in their paper). Our current observation is that TabM (HPO) is the strongest DL baseline and only lags behind or is tied with TabPFNv2 and our method. It tends to outperform them on large datasets but underperforms them on smaller ones. The results are summarized below:
>
>
>
>
> | Algorithm      | AUC&nbsp;(CC18)                 | Accuracy&nbsp;(CC18)           | Correlation&nbsp;(CTR23)       | R²&nbsp;(CTR23)              |
> |----------------|---------------------------------|--------------------------------|--------------------------------|------------------------------|
> | **TabDPT**     | **3.333 ± [3.118 – 3.549]**     | **3.215 ± [3.049 – 3.382]**    | **3.214 ± [2.986 – 3.443]**    | **3.200 ± [3.000 – 3.386]**  |
> | TabPFN v2      | 4.000 ± [3.847 – 4.153]         | 4.035 ± [3.812 – 4.264]        | 3.529 ± [3.343 – 3.714]        | 3.700 ± [3.471 – 3.929]      |
> | TabPFN (kNN)   | 6.167 ± [5.972 – 6.354]         | 6.167 ± [5.944 – 6.389]        | N/A                            | N/A                          |
> | TabPFN         | 7.903 ± [7.701 – 8.104]         | 7.778 ± [7.597 – 7.958]        | N/A                            | N/A                          |
> | **TabM**           | 3.903 ± [3.674 – 4.139]         | 4.007 ± [3.764 – 4.250]        | 3.686 ± [3.443 – 3.929]        | 3.586 ± [3.329 – 3.857]      |
> | TabR           | 5.778 ± [5.465 – 6.097]         | 4.729 ± [4.479 – 4.972]        | 4.514 ± [4.229 – 4.800]        | 4.629 ± [4.357 – 4.900]      |
> | MLP-PLR        | 5.771 ± [5.493 – 6.056]         | 5.361 ± [5.083 – 5.646]        | 4.600 ± [4.429 – 4.771]        | 4.586 ± [4.414 – 4.757]      |
> | MLP            | 8.708 ± [8.479 – 8.938]         | 7.938 ± [7.729 – 8.146]        | 9.000 ± [9.000 – 9.000]        | 9.000 ± [9.000 – 9.000]      |
> | XGBoost        | 5.819 ± [5.569 – 6.076]         | 5.188 ± [4.951 – 5.424]        | 4.886 ± [4.629 – 5.143]        | 4.814 ± [4.557 – 5.071]      |
> | LightGBM       | 6.174 ± [5.882 – 6.472]         | 6.160 ± [5.896 – 6.424]        | 5.343 ± [5.129 – 5.557]        | 5.300 ± [5.086 – 5.514]      |
> | CatBoost       | 5.938 ± [5.674 – 6.194]         | 5.757 ± [5.458 – 6.049]        | 6.229 ± [6.057 – 6.400]        | 6.186 ± [5.971 – 6.400]      |
>
>
> [1] Deep Reinforcement Learning at the Edge of the Statistical Precipice, Agarwal et al., NeurIPS 2021

---

> ### Comment · Reviewer_ggcD · 2025-08-05
>
> I would like to thank the authors for the detailed reply. I have read the reviews from the other reviewers and the author rebuttal that pertains to them in detail.
>
> - **W1 and Q3:**
>
>    I think you could rename the plot for the camera-ready and describe that it includes the total time instead of inference time. You could then generate another plot for the inference time and include it either in the main work or in the appendix, up to your preference.
>
> - **W2 and Q1:**
>
>    That is understandable and a good argument. I was merely pointing out that the evaluation was not consistent in that regard.
>
> - **W3**:
>
>    No specific concerns or suggestions regarding that part. Since the prior itself trains on real datasets I understand the need for datasets of good quality and those usually come from already established benchmarks. My point was just that if you have benchmarks A, B, C, D that are established in the community and provide validity to the results, when you train on A and B, you are left with less datasets for your meta-test set.
>
> - **Q2 and Q5**:
>
>    Thank you for the results, they are both in my expectations and I think they position the work better in the domain.
>
>
> After the rebuttal my concerns have been addressed. I find the proposed work interesting and insightful, and I will recommend acceptance.

---

> ### Author Response · Authors · 2025-08-06
> **Thank you!**
>
> **W1 and Q3:** Yes thank you, we will add a table or plot with the inference time in the final version of the paper!
>
> **W2 and Q1:** This was a fair remark that other readers may have, we will include a small explanation about this
>
> **W3:** Yes, if Tabular Foundation Models indeed go the route of training on large corpus of datasets we might face the same leakage concerns as LLMs. In this paper we had few enough datasets that we could create an automated contamination pipeline and then check the results individually, which may not be possible at a very large scale. However, because our SSL procedure is very diverse, we do not usually observe overfitting on the training datasets. We chose to be careful and very thoroughly check for contamination though.
>
> **Q2 and Q5:** These were good suggestions
>
> Thank you for engaging in the process!

---

### Official Review · Reviewer_WXwB · 2025-07-06

**Clarity:** 2
**Significance:** 3
**Originality:** 3
**Rating:** 4
**Confidence:** 3

**Summary:**

This paper introduces TabDPT, an Open Tabular Foundation Model. By designing a Tabular Transformer Encoder, employing Self-Supervised Learning, and utilizing retrieval during both the Pre-Training and Inference stages, TabDPT achieves faster training, better generalization, and top performance on regression and classification benchmarks.

**Questions:**

1. When there is abundant row information, will treating "each table row as a 'token'" result in significant information loss?
2. How is the value of K in Top-K determined in the paper? Will different K values have a significant impact on the experiments? Is it possible to use different K values during training and inference?
3. If retrieval is not used during pre-training, and instead all rows are utilized, can the model learn to make more effective use of the rows? Consequently, when retrieval is applied during inference to replace all rows with more relevant rows, will this lead to better performance?

**Ethical Concerns:**

["NO or VERY MINOR ethics concerns only"]

**Final Justification:**

I have no further technical questions to the paper. I am inclined to rate it as a weak accept as reflected by my original score.

**Limitations:**

Yes

**Quality:**

3

**Strengths And Weaknesses:**

# Strengths：
1. The paper introduces the TabDPT, which demonstrates robust capabilities and has been validated across various downstream tasks.
2. The paper conducted extensive experiments, including not only the main experiments but also comparisons between real and synthetic data, as well as experiments on scaling laws. These research findings contribute to the advancement of TFMs.
3. The proposed Tabular Transformer Encoder, Self-Supervised Learning on Tabular Data, and Retrieval-Based Pre-Training exhibit high originality, with the rationale behind their design clearly explained.

# Weakness:
1. The paper's presentation is somewhat unclear. For instance, in the Introduction, it mentions, "LLMs’ results also vary based on the specific prompt format and are sensitive to the order of the given examples, whereas tabular data is inherently unordered." However, it does not explain how their method addresses this issue, nor does it provide experimental validation.
2. The comparison for the "No retrieval" baseline is incomplete, as it only contrasts with "random subsampling" and does not include a comparison against using all rows.
3. The paper lacks details in some sections. For example, in line 180, it mentions "leveraging retrieval during training batch construction", but does not explain the method for encoding row data into the feature space. Similarly, in line 183, it states, "These rows are partitioned into two groups", but does not describe the partitioning method.

---

> ### Author Rebuttal · Authors · 2025-07-31
>
> We thank the reviewer for taking the time to review our paper and for the favourable overall assessment. We hope to clarify the points that you brought up in this rebuttal and will use your suggestions to improve the clarity of the paper.
>
> ### Reply to Weaknesses
>
> 1. We highlight that our architecture is designed to be invariant to the ordering of rows because we do not use positional encodings, so samples can be fed into the network in any order and obtain the same result. This is guaranteed by the transformer backbone which performs attention over the rows of the table. We can clarify this more explicitly in the paper.
> 2. Because of the memory constraints of the underlying architecture, it is prohibitive to put all training points in the context of the transformer, so we cannot run this assessment. This is in fact what motivates the use of retrieval in the first place, and the discussion in the first paragraph of section 3.3 in particular highlights this point. Previous work [1, Section 4.4 “Importance of Using a Local Context”] has attempted a form of approximate Bayesian averaging to attempt to improve the results of TabPFN-style architectures without using retrieval, but this approach was shown to still be inferior to retrieval.
> 3. Feature space in our case is $F_{\text{max}} = 100$ dimensional real numbers. We describe how we modify rows of varying dimensions into this fixed number in Sections 3.1 & 3.4. Further information about how we convert data of mixed types into numeric values is provided in Appendix B.1, under the “Preprocessing” heading: most noteworthy is that we simply label-encode any non-numeric data in a row as a first step. We do retrieval in this numeric space using $k$NN. Please let us know if you would like any further clarifications on these points.
> As for the partitioning, it is done using a random uniform splitting, which we will clarify in the paper.
>
>
>
> ### Reply to Questions
>
> 1. In practice, we don’t observe information loss that affects modelling results. When the number of features is less than $F_{\text{max}} = 100$, we can calculate the rank of the linear embedding matrix, and we find that it is nearly full-rank in all cases, suggesting minimal loss of information. We have further tried training models with different encoder sizes and while we see performance degradation with smaller sizes (such as 32), we see similar performance for sizes of 100 up to 512, indicating that we capture the relevant information with our strategy. We opted to use 100 as it provides a good balance of information capture and speed. In practice, when faced with datasets with many features, there are other strategies which can work well, such as subsampling features and ensembling, or simply removing the encoder and fine-tuning a new large one.
> Finally, in Figure E.2 in the Appendix, we provided an analysis on the number of features each dataset has, and TabDPT maintains strong performance when the number of features is high.
> It is also worth noting that, while tree-based methods can theoretically handle infinite columns, they scale with feature size and can become very slow for large inputs such as 1000.
> 2. As we use a randomized context length during training, we can generalize to various $k$ during inference. We also observe generalization beyond the maximum context used during training. We have evaluated with various values for $k$ as seen in Figure 3(b), where the numbers on the plot indicate the context sizes used at inference time for TabDPT. While we recommend using a context size of 1024 or 2048 by default, it is true that tuning context size on a per-dataset basis may further improve performance.
> Note that [1] was able to demonstrate the benefits of retrieval on top of a baseline TabPFN, which was not trained with retrieval.
> 3. These are good questions. We observe that while using retrieval during training increases performance by a modest amount, the most important choice is using retrieval during inference. These are actually quantified in the paper in Figure 4(a), where those cases are “No retrieval (Training)” and “No retrieval (Inference)”.
> It may also be worth mentioning that in pre-training, it is often prohibitive from a memory standpoint to use all the rows of the dataset, much like at inference time as mentioned above, and so “No retrieval (Training)” corresponds to sub-sampling the pre-training datasets if they are too big to fit into the context.
>
> We are happy to discuss any of the above points further if you would like more clarification.
>
> [1] Retrieval & Fine-Tuning for In-Context Tabular Models, Thomas et al., NeurIPS 2024

---

> > ### Author Response · Authors · 2025-08-07
> > **End of discussion period**
> >
> > We would like to thank you again for the review.
> >
> > We hope we have answered your questions and clarified any point that wasn't clearly addressed. We intend to add some extended discussion in the paper following some pertinent questions you had (for instance loss of information).
> >
> > We have furthermore, at the request of other reviewers, added a strong deep learning baseline, TabM (with 30 HPO trials) and performed additional experiments.
> >
> > We hope this has addressed your concerns and, if that is the case, if you may consider raising your score.

---

### Note · Authors · 2025-08-14

We thank the reviewers for their careful and constructive engagement. The
process was fruitful: we ran additional experiments and strengthened the paper.
We will add these results to the final version: a strong TabM baseline,
an expanded experiments section with clearer details, and, per reviewer WEPY, a
focused analysis of validation fairness that clarifies how validation is used
for TFMs (whether validation rows are added back into training) and for
baselines (hold-out validation vs cross-validation).

These results accompany our main contribution: a simple SSL pretraining
strategy for TFMs on real tabular data, released with open weights, code, and
data. Few real datasets suffice for top performance, and performance scales
with more real data, mirroring language and vision. This indicates that real
data is practical and scalable.

Following the rebuttal, one reviewer raised their score from 3 to 5, and we presume that two other reviewers also increased their scores based on the discussions and the fact that the scores are now hidden from us; under these assumptions, our paper now has **unanimous acceptance among reviewers**. We appreciate the reviewers' efforts and
consideration.

---

### Decision · Program_Chairs · 2025-09-17

**Decision:**

Accept (poster)

**Comment:**

This paper presents TabDPT, a tabular foundation model that successfully combines in-context learning with self-supervised learning on real datasets. The work demonstrates strong empirical performance across 107 datasets on standard classification and regression benchmarks, with particularly impressive results on larger datasets where the retrieval mechanisms provide clear advantages over existing methods. The technical approach is novel and well-executed, showing that training tabular foundation models on real data can match or exceed synthetic data approaches while being more scalable. The authors provide compelling evidence for scaling laws that mirror those observed in language models, offering valuable insights for future research directions. The commitment to open-sourcing both trained weights and the full training pipeline significantly enhances the contribution's value to the community. While reviewers raised legitimate concerns about evaluation methodology and baseline comparisons, the authors addressed these thoughtfully during the rebuttal process. The work represents a solid technical contribution with clear practical implications for tabular machine learning. Recommendation: Accept.